# HYPERGRAPH DYNAMIC SYSTEM

**Jielong Yan**[1], **Yifan Feng**[1], **Shihui Ying**[2] **& Yue Gao**[1*]
[1]School of Software, BNRist, THUIBCS, BLBCI, Tsinghua University
[2]Department of Mathematics, School of Science, Shanghai University
{yanjl.jason,evanfeng97}@gmail.com, shying@shu.edu.cn
gaoyue@tsinghua.edu.cn

## ABSTRACT

Recently, hypergraph neural networks (HGNNs) exhibit the potential to tackle tasks with high-order correlations and have achieved success in many tasks. However, existing evolution on the hypergraph has poor controllability and lacks sufficient theoretical support (like dynamic systems), thus yielding sub-optimal performance. One typical scenario is that only one or two layers of HGNNs can achieve good results and more layers lead to degeneration of performance. Under such circumstances, it is important to increase the controllability of HGNNs. In this paper, we first introduce hypergraph dynamic systems (HDS), which bridge hypergraphs and dynamic systems and characterize the continuous dynamics of representations. We then propose a control-diffusion hypergraph dynamic system by an ordinary differential equation (ODE). We design a multi-layer $HDS^{ode}$ as a neural implementation, which contains control steps and diffusion steps. $HDS^{ode}$ has the properties of controllability and stabilization and is allowed to capture long-range correlations among vertices. Experiments on 9 datasets demonstrate $HDS^{ode}$ beat all compared methods. $HDS^{ode}$ achieves stable performance with increased layers and solves the poor controllability of HGNNs. We also provide the feature visualization of the evolutionary process to demonstrate the controllability and stabilization of $HDS^{ode}$.

## 1 INTRODUCTION

Real-world correlation data inherently includes high-order correlations that graphs cannot fully depict, such as group ties in social networks (Bu et al., 2010) and co-actor relationships in movies (Fan et al., 2021). The hyperedge in hypergraphs can connect two or more vertices, allowing the hypergraph to perform high-order correlation modeling compared to the graph. Recently, hypergraph neural networks have gained interest due to their ability to handle high-order correlation tasks such as drug-target interactions (Ruan et al., 2021), social recommendation (Xia et al., 2021), and gene expression imputation (Viñas et al., 2023). The information propagation from layer to layer in traditional convolutional neural networks can be regarded as discrete information diffusion (Lin et al., 2017; Saharia et al., 2022; Rombach et al., 2022), and as the layer goes deeper (more diffusion steps), the expressive ability of features also increases (Rolnick & Tegmark, 2017; Li et al., 2021).

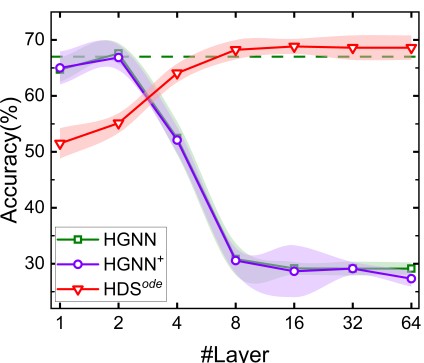

Figure 1: Performance comparison of different methods with respect to varied numbers of neural network layers on Cora-CA dataset.

However, we found that existing hypergraph neural networks tolerate only small diffusion steps (*e.g.*, HGNN (Feng et al., 2019) contains only 2 layers), while the performance drops significantly by raising layer numbers, as shown in Figure 1. The reason is that diffusion in hypergraph neural networks is merely simple message-smoothing within neighbors, resulting in poor controllability

---

*Corresponding author: Yue Gao

and stabilization with controllability referring to the capability to fine-tune and adjust the diffusion process. Graph ODE-based methods bridge graph evolution and dynamic systems to the smoother representation of diffusion dynamics, enabling deeper networks (Poli et al., 2019; Xhonneux et al., 2020; Rusch et al., 2022). However, it is challenging to directly apply these methods to high-order structures since diffusion in pair-wise correlation structures (graphs) and beyond pair-wise correlation structures (hypergraphs) follow different paradigms. In this paper, we aim to propose a hypergraph dynamic system to improve the controllability and stabilization of information diffusion on the hypergraph, thereby improving the expressive ability of features as the number of diffusion steps increases.

In this paper, we theoretically introduce hypergraph dynamic systems, which bridge hypergraphs and dynamic systems. We propose a specific hypergraph dynamic system based on a control-diffusion ODE. Based on this, we propose a neural network implementation $\text{HDS}^{ode}$ that achieves controllable and stable long-distance diffusion on hypergraphs. Our $\text{HDS}^{ode}$ method exhibits steady performance as the diffusion steps (layers) increase, shown in Figure 1. We also present the properties of $\text{HDS}^{ode}$, including stability analysis and the connection to hypergraph neural networks. In our experiments, we employ 9 real-world hypergraph benchmarks and thoroughly evaluate $\text{HDS}^{ode}$ in an inductive setting and a production setting with 8 compared methods to validate the effectiveness of $\text{HDS}^{ode}$. Furthermore, we provide feature visualizations of the evolutionary process to demonstrate the controllability and stability of $\text{HDS}^{ode}$. We summarize our contributions as follows:

- We introduce hypergraph dynamic systems to establish the connection between hypergraph and dynamic systems. This dynamic system characterizes dynamic continuous representations. We then propose a control-diffusion hypergraph dynamic system based on an ODE.

- We design a multi-layer framework $\text{HDS}^{ode}$ as a neural implementation of the hypergraph dynamic system to generate accurate vertex representations and prove the properties of $\text{HDS}^{ode}$, including stability analysis which indicates that $\text{HDS}^{ode}$ can capture long-range relations among vertices.

- We perform an extensive empirical evaluation of $\text{HDS}^{ode}$ on 9 datasets, indicating that $\text{HDS}^{ode}$ can achieve best performance compared with all methods. Moreover, $\text{HDS}^{ode}$ can achieve stable performance with respect to the increase of 16 or more layers, which solves the poor controllability issue of HGNNs.

## 2   RELATED WORKS

**Hypergraph neural networks.**   Hypergraph neural networks have been proposed for convolution operations on hypergraphs to handle non-Euclidean hypergraph data, which is first introduced from the spectral perspective by HGNN (Feng et al., 2019). Hyper-Atten (Bai et al., 2021) additionally focuses on the hypergraph attention module based on HGNN. Besides, HyperGCN (Yadati et al., 2019) is proposed for training GCN on hypergraphs by converting hypergraphs into graphs with intermediaries to represent hyperedges. In addition to the spectral-based methods mentioned above, $\text{HGNN}^+$ (Gao et al., 2022) provides a spatial-based method for propagating messages from vertices to hyperedges and then to vertices. UniGNN (Huang & Yang, 2021) presents a unified structure for message passing in graph and hypergraph neural networks, allowing common graph neural network models (*e.g.*, GCN (Kipf & Welling, 2017), GAT (Veličković et al., 2018), GIN (Xu et al., 2018), and GraphSAGE (Hamilton et al., 2017)) to be generalized to hypergraphs.

**Neural ordinary differential equations (Neural ODEs).**   Neural ODE is first proposed by Chen et al. (2018) to represent the continuous dynamics of the hidden representations. ODEs parameterized by neural networks have been utilized in recent years to analyze structured graph data and develop connections between dynamic systems and correlation structures. GDE (Poli et al., 2019) is a continuous deep correspondence formal extension of graph neural networks. CGNN (Xhonneux et al., 2020) characterizes the continuous dynamics of vertex representations in terms of solutions to linear graph diffusion differential equations. In addition, GREAD (Choi et al., 2022) adds a reaction term to the graph diffusion ODE to obtain sharpening of the vertex representation, and GraphCON (Rusch et al., 2022) models control and damping oscillators and couples them based on the graph structure. Further, there are also implementations based on partial differential equations to model

deep learning on the correlation structure as a continuous diffusion process (Chamberlain et al., 2021; Thorpe et al., 2021; Bodnar et al., 2022; Eliasof et al., 2021).

## 3 PRELIMINARY

**Notations and problem statement.** Compared to the simple graph, each hyperedge in the hypergraph is a subset of the vertex set. Generally, a hypergraph is defined as $\mathcal{G} = (\mathcal{V}, \mathcal{E})$ with $\mathcal{V}, \mathcal{E}$ representing the vertex set and hyperedge set, respectively. The hyperedges is denoted by an incidence matrix $\boldsymbol{H} \in \{0,1\}^{|\mathcal{V}| \times |\mathcal{E}|}$, whose entities are defined as $H_{v,e} = \mathbf{1}(v \in e)$ with indicator function $\mathbf{1}(\cdot)$. The degree of a vertex $v \in \mathcal{V}$ is defined as $d(v) = \sum_{e \in \mathcal{E}} H_{v,e}$. Similarly, the degree of hyperedge $e \in \mathcal{E}$ is defined as $\delta(e) = \sum_{v \in \mathcal{V}} H_{v,e}$. The diagonal degree matrices of vertex and hyperedge are denoted by $\boldsymbol{D}_v = \mathrm{diag}(\boldsymbol{d})$ and $\boldsymbol{D}_e = \mathrm{diag}(\boldsymbol{\delta})$, respectively. Given a hypergraph $\mathcal{G}$, a corresponding vertex feature matrix $\boldsymbol{Z}_v \in \mathbb{R}^{|\mathcal{V}| \times c}$, and a corresponding hyperedge feature matrix $\boldsymbol{Z}_e \in \mathbb{R}^{|\mathcal{E}| \times c}$, our goal is to learn a vertex representation $\boldsymbol{Y}_v$ and a hyperedge representation $\boldsymbol{Y}_e$.

**Hypergraph neural networks.** Most current hypergraph neural networks follow the message-passing framework. In each layer, the input vertex representations are first aggregated into hyperedges, and then the output vertex representations are obtained from the corresponding hyperedges. Formally, in the $k$-th layer, the output vertex representations $\boldsymbol{x}_v^{(k)}$ are obtained from the previous representations using aggregation function AGG and update function UPD as:

$$\boldsymbol{x}_v^{(k)} = \mathrm{UPD}(\boldsymbol{x}_v^{(k-1)}, \mathrm{AGG}(\{\boldsymbol{x}_e^{(k)} : e \in \mathcal{N}_e(v)\})), \quad \boldsymbol{x}_e^{(k)} = \mathrm{AGG}(\{\boldsymbol{x}_v^{(k-1)} : v \in \mathcal{N}_v(e)\}), \quad (1)$$

where the $\mathcal{N}_e(v)$ and $\mathcal{N}_v(e)$ are the vertex and hyperedge neighbor function, respectively.

**Neural ordinary differential equations.** For certain types of models, including residual networks (He et al., 2016), the conversion of the hidden feature is regarded as the following discrete system: $\boldsymbol{x}(t+1) = \boldsymbol{x}(t) + f(\boldsymbol{x}(t), \theta(t))$. It can be considered as the forward Euler discretization form of the following first-order ODE equation with time step $\Delta t = 1$ as $\frac{d\boldsymbol{x}}{dt} = f(\boldsymbol{x}(t), \theta)$, where $f$ is a parameterized function defined in a dynamic system. In the remainder of the paper, $\frac{d\boldsymbol{x}}{dt}$ will be abbreviated as $\dot{\boldsymbol{x}}$ for the sake of brevity.

## 4 METHOD

In this section, we first theoretically introduce hypergraph dynamic systems. Then, we provide a specific hypergraph dynamic system form based on an ODE. Next, we divide the ODE into a control step and a diffusion step by an ODE discretization for neural implementation. Furthermore, we describe the detailed neural implementation of the HDS$^{ode}$ framework, and also the time complexity of the control step and the diffusion step.

### 4.1 HYPERGRAPH DYNAMIC SYSTEM.

We first propose the definition of hypergraph dynamic systems based on the following equation:

$$\begin{bmatrix} \dot{\boldsymbol{X}}_v \\ \dot{\boldsymbol{X}}_e \end{bmatrix} = f\left(\begin{bmatrix} \boldsymbol{X}_v(t) \\ \boldsymbol{X}_e(t) \end{bmatrix}\right) \quad \text{and} \quad \begin{bmatrix} \boldsymbol{X}_v(0) \\ \boldsymbol{X}_e(0) \end{bmatrix} = \begin{bmatrix} \boldsymbol{Z}_v \\ \boldsymbol{Z}_e \end{bmatrix}, \quad (2)$$

where $\boldsymbol{X}_v(t)$ and $\boldsymbol{X}_e(t)$ represent the vertex representation matrix and the hyperedge representation matrix at time $t$, respectively. The function $f$ represents the velocity of representation in the dynamic system, $\boldsymbol{Z}_v$ and $\boldsymbol{Z}_e$ represent the initial conditions of vertex features and hyperedge features, respectively. Due to the continuous existence of timestamp $t$, the above hypergraph dynamic systems can produce the representation status at any moment.

**ODE-based hypergraph dynamic system.** The velocity function $f$ in equation 2 can be described in different ways. We consider the velocity function as a union of a control function and a diffusion function to propose an ODE-based hypergraph dynamic system as follows:

$$\begin{bmatrix} \dot{\boldsymbol{X}}_v \\ \dot{\boldsymbol{X}}_e \end{bmatrix} = \begin{bmatrix} g_v(\boldsymbol{X}_v(t)) \\ g_e(\boldsymbol{X}_e(t)) \end{bmatrix} + \boldsymbol{A} \begin{bmatrix} \boldsymbol{X}_v(t) \\ \boldsymbol{X}_e(t) \end{bmatrix}. \quad (3)$$

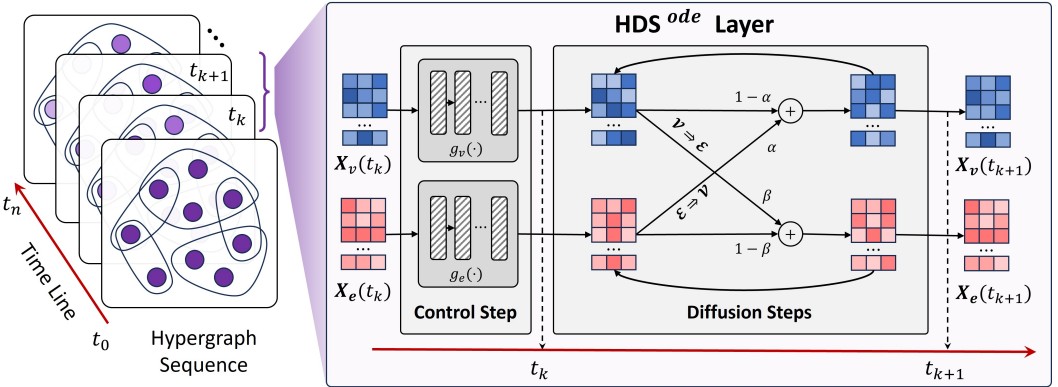

Figure 2: Illustration of our HDS$^{ode}$ framework.

Here, in the first term, $g_v$ and $g_e$ are the control functions, acting as the control velocity of each vertex representation and hyperedge representation, respectively. The second term is the diffusion term, where $\boldsymbol{A}$ denotes the diffusion velocity effect between the vertex representation and the hyperedge representation in the dynamic system by the correlation of the hypergraph. The diffusion term describes the process by which features or representations teleport across the vertices and hyperedges of the hypergraph. The control term specifically refers to a fine-tuning step that complements the primary diffusion process and acts as an auxiliary function, adjusting and controlling the diffusion term to align more precisely with the downstream goals.

$$\begin{bmatrix} \boldsymbol{X}_v(T) \\ \boldsymbol{X}_e(T) \end{bmatrix} = \begin{bmatrix} \boldsymbol{X}_v(0) \\ \boldsymbol{X}_e(0) \end{bmatrix} + \int_0^T f\left( \begin{bmatrix} \boldsymbol{X}_v(t) \\ \boldsymbol{X}_e(t) \end{bmatrix} \right) dt. \tag{4}$$

Given vertex features $\boldsymbol{X}_v(0)$ and hyperedge features $\boldsymbol{X}_e(0)$ as input, the vertex representations $\boldsymbol{X}_v(T)$ and hyperedge representations $\boldsymbol{X}_e(T)$ of time $T$ are generated using the integral for the learning tasks in the hypergraph as equation 4. Not only can we obtain accurate final representations, but more importantly, we can also acquire the dynamic changes of representations from $\boldsymbol{X}_v(0), \boldsymbol{X}_e(0)$ to $\boldsymbol{X}_v(T), \boldsymbol{X}_e(T)$ by changing the upper bound of the integral.

**ODE discretization with Lie-Trotter splitting.** We expect to propose a multi-layer neural network framework HDS$^{ode}$ related to the above ODE-based hypergraph dynamic system to obtain accurate representations. We first employ the Lie-Trotter (Geiser, 2009) splitting method for the discretization of equation 3, which is as follows:

$$\begin{bmatrix} \boldsymbol{X}_v(t+\frac{1}{2}) \\ \boldsymbol{X}_e(t+\frac{1}{2}) \end{bmatrix} = \begin{bmatrix} \boldsymbol{X}_v(t) \\ \boldsymbol{X}_e(t) \end{bmatrix} + \begin{bmatrix} g_v(\boldsymbol{X}_v(t)) \\ g_e(\boldsymbol{X}_e(t)) \end{bmatrix}, \begin{bmatrix} \boldsymbol{X}_v(t+1) \\ \boldsymbol{X}_e(t+1) \end{bmatrix} = \begin{bmatrix} \boldsymbol{X}_v(t+\frac{1}{2}) \\ \boldsymbol{X}_e(t+\frac{1}{2}) \end{bmatrix} + \boldsymbol{A} \begin{bmatrix} \boldsymbol{X}_v(t+\frac{1}{2}) \\ \boldsymbol{X}_e(t+\frac{1}{2}) \end{bmatrix}, \tag{5}$$

where the time step is integrated into control functions $g_v, g_e$, and diffusion matrix $\boldsymbol{A}$. We notice that each iteration of representations in the dynamic system has a time interval of 1 and time iteration is separated into a control step and a diffusion step. Similar to the residual network, the control step modifies the representation of vertices and hyperedges by the control functions. The diffusion step propagates representation messages between vertices and hyperedges according to matrix $\boldsymbol{A}$. It is worth mentioning that the diffusion step is parameter-free. If the diffusion matrix $\boldsymbol{A}$ is suitably designed, representations are stable to a specific value in the diffusion step with proof in Section D.

## 4.2 HDS$^{ode}$: Neural Implementation of ODE-Based Hypergraph Dynamic System

In the following, we present the implementation of HDS$^{ode}$ framework to the previous analysis by introducing the neural implementation of the control step and diffusion step in the HDS$^{ode}$ layer, respectively. Furthermore, we will analyze the time complexity of the two steps, respectively. The illustration of the HDS$^{ode}$ framework can be found in Figure 2.

**Neural implementation of control step.** Our HDS$^{ode}$ layer allows any function with the same input and output dimensions as control functions. In this paper, we take two simple one-layer fully

connected networks as modifications of vertex representations and hyperedge representations, respectively. Specifically, it can be expressed by the following formula:

$$\begin{bmatrix} \boldsymbol{X}_v(t+\frac{1}{2}) \\ \boldsymbol{X}_e(t+\frac{1}{2}) \end{bmatrix} = \begin{bmatrix} \boldsymbol{X}_v(t) \\ \boldsymbol{X}_e(t) \end{bmatrix} + \sigma\left(\begin{bmatrix} \boldsymbol{W}_v\boldsymbol{X}_v(t)) + \boldsymbol{b}_v \\ \boldsymbol{W}_e\boldsymbol{X}_e(t)) + \boldsymbol{b}_e \end{bmatrix}\right), \tag{6}$$

where $\sigma$ is the activate function, $\boldsymbol{W}_v, \boldsymbol{W}_e \in \mathbb{R}^{c \times c}$ are the learnable weight matrices of vertex representations and hypergraph representations, respectively, and $\boldsymbol{b}_v, \boldsymbol{b}_e \in \mathbb{R}^c$ are learnable biases.

**Neural implementation of diffusion step.** The design of the diffusion matrix $\boldsymbol{A}$ is essential during the diffusion process. If not suitably created, vertex representations and hyperedge representations will diverge and become uncontrollable. In this study, we provide a design that holds both stability and interpretability, as follows:

$$\begin{bmatrix} \boldsymbol{X}_v(t+1) \\ \boldsymbol{X}_e(t+1) \end{bmatrix} = \begin{bmatrix} \boldsymbol{X}_v(t+\frac{1}{2}) \\ \boldsymbol{X}_e(t+\frac{1}{2}) \end{bmatrix} + \boldsymbol{A}\begin{bmatrix} \boldsymbol{X}_v(t+\frac{1}{2}) \\ \boldsymbol{X}_e(t+\frac{1}{2}) \end{bmatrix} \quad \text{and} \quad \boldsymbol{A} = \begin{bmatrix} -\alpha_v\boldsymbol{I} & \alpha_v\boldsymbol{D}_v^{-1}\boldsymbol{H} \\ \alpha_e\boldsymbol{D}_e^{-1}\boldsymbol{H}^\top & -\alpha_e\boldsymbol{I} \end{bmatrix}, \tag{7}$$

where $\alpha_v$ and $\alpha_e$ are hyperparameters representing the teleport probabilities of vertices and hyperedges, respectively. We furthermore expand the matrix multiplication term to obtain the vertex representation as $\boldsymbol{X}_v(t+1) = (1-\alpha_v)\boldsymbol{X}_v(t+\frac{1}{2}) + \alpha_v\boldsymbol{D}_v^{-1}\boldsymbol{H}\boldsymbol{X}_e(t+\frac{1}{2})$. The first term denotes that the vertex representations stay unmodified with a keep-rate $1-\alpha_v$ in the diffusion process. The second term denotes that the representation of the hyperedges directly connected to each vertex is aggregated by average with a contribution proportion of $\alpha_v$, where $\boldsymbol{H}\boldsymbol{X}_e(t+\frac{1}{2})$ represents vertex-level aggregation, and $\boldsymbol{D}_v^{-1}$ represents an average normalization matrix. Similarly, the hyperedge representation can be calculated as $\boldsymbol{X}_e(t+1) = \alpha_e\boldsymbol{D}_e^{-1}\boldsymbol{H}^\top\boldsymbol{X}_v(t+\frac{1}{2}) + (1-\alpha_e)\boldsymbol{X}_e(t+\frac{1}{2})$, where the first and the second term represent the aggregation from the vertex representation at $\alpha_e$ rate and the original representation retained at $1-\alpha_e$ rate, respectively.

We obtain the vertex and hyperedge representations of any non-negative integer timestamp $t$ using the implementation described above. The hypergraphs from time $t_0$ to time $t_n$ constitute a hypergraph sequence as the left part in Figure 2, which corresponds to the evolution process of the hypergraph dynamic system. Once the time $T$ of the hypergraph dynamic system is selected, the final vertex representation is $\boldsymbol{Y}_v = \boldsymbol{X}_v(T)$, and the hyperedge representation is $\boldsymbol{Y}_e = \boldsymbol{X}_e(T)$. The whole algorithm of HDS$^{ode}$ is shown in Appendix A.

**Time complexity analysis.** Here, we analyze the time complexity of the control step and the diffusion step in each HDS$^{ode}$ layer. In the control step, the running time is limited by multiplying the weight matrices and the representations, so the time complexity of the control step is $O((|\mathcal{V}| + |\mathcal{E}|)c^2)$, where $c$ denotes the dimension of representations. In the diffusion step, the running time is limited by the matrix multiplication operations of representations aggregation (*i.e.*, $\boldsymbol{H}\boldsymbol{X}_e(t+\frac{1}{2})$ and $\boldsymbol{H}^\top\boldsymbol{X}_v(t+\frac{1}{2})$). Considering that the incidence matrix $\boldsymbol{H}$ is a sparse matrix, the time complexity is $O((tr(\boldsymbol{D}_v) + tr(\boldsymbol{D}_e))c)$. It should be noticed that the time complexity of the control term is quadratic concerning the representation dimension, where the diffusion term is linear. Therefore, in the implementation, we mask the control function in most time iterations to lower the total running duration of the framework (*i.e.*, a control step is conducted every certain number of layers).

## 5 PROPERTIES OF HDS$^{ode}$

In this section, we introduce the properties of our proposed HDS$^{ode}$. First, we provide the eigenvalue propositions of the diffusion matrix $\boldsymbol{A}$ in HDS$^{ode}$ and then explore the stability of diffusion steps. Then, we discuss the relationship between HDS$^{ode}$ and hypergraph neural networks.

### 5.1 STABILITY ANALYSIS.

Since the diffusion step in the ODE reflects the time iteration of the vertex and hyperedge representations, the stability analysis of diffusion is essential. We first analyze the proposition of the eigenvalue of the diffusion matrix $\boldsymbol{A}$. Then, we perform a stability analysis on the diffusion process.

**Proposition 5.1.** *Assume that the diffusion matrix's eigendecomposition is $\boldsymbol{A} = \boldsymbol{U}\boldsymbol{\Lambda}\boldsymbol{U}^{-1}$ with eigenvalue matrix $\boldsymbol{\Lambda} = diag(\lambda_i)$ and eigenvectors $\boldsymbol{u}_i$ in equation 7, the eigenvalue $\lambda_i$ lies in the left half-plane of the complex plane or is $0$.*

Table 1: Test accuracy (%) and standard deviation of semi-supervised vertex classification on a transductive setting. "OOM" and "Avg. rank" represent "out of memory" and "Average rank", respectively. The best results are shown in bold.

| Model | Cora-CA | DBLP-CA | News20 | IMDB4k-CA | IMDB4k-CD | DBLP4k-CC | DBLP4k-CP | Avg. rank |
|---|---|---|---|---|---|---|---|---|
| GCN | $65.99_{\pm3.69}$ | $82.22_{\pm1.05}$ | $67.57_{\pm0.70}$ | $43.47_{\pm2.39}$ | $41.02_{\pm2.22}$ | $90.18_{\pm1.22}$ | $64.47_{\pm0.90}$ | 7.6 |
| GraphSAGE | $66.44_{\pm2.82}$ | $81.07_{\pm1.50}$ | $69.59_{\pm0.89}$ | $42.05_{\pm1.95}$ | $41.07_{\pm2.11}$ | $92.18_{\pm0.38}$ | $64.34_{\pm1.58}$ | 8.0 |
| GDE | $66.01_{\pm1.02}$ | $82.61_{\pm1.74}$ | $69.95_{\pm0.41}$ | $43.95_{\pm2.64}$ | $41.80_{\pm0.98}$ | $92.45_{\pm0.45}$ | $67.71_{\pm2.46}$ | 4.9 |
| GraphCON | $66.72_{\pm1.71}$ | $82.06_{\pm1.11}$ | OOM | $43.94_{\pm2.36}$ | $41.92_{\pm2.89}$ | OOM | $67.94_{\pm1.04}$ | 4.4 |
| HGNN | $67.58_{\pm1.83}$ | $82.83_{\pm1.09}$ | $76.58_{\pm0.94}$ | $43.21_{\pm2.39}$ | $41.08_{\pm2.43}$ | $93.46_{\pm0.77}$ | $67.99_{\pm2.12}$ | 4.0 |
| HGNN$^+$ | $66.85_{\pm2.24}$ | $82.40_{\pm1.27}$ | $76.49_{\pm1.30}$ | $43.74_{\pm1.42}$ | $41.49_{\pm2.54}$ | $93.46_{\pm1.09}$ | $68.76_{\pm2.73}$ | 3.7 |
| UniGCN | $66.47_{\pm2.04}$ | $82.36_{\pm1.09}$ | $76.56_{\pm1.21}$ | $43.34_{\pm3.26}$ | $41.33_{\pm2.50}$ | $93.28_{\pm0.87}$ | $67.68_{\pm1.90}$ | 5.4 |
| UniSAGE | $68.59_{\pm1.61}$ | $82.16_{\pm1.25}$ | $75.52_{\pm1.22}$ | $42.82_{\pm2.66}$ | $41.62_{\pm3.05}$ | $93.64_{\pm0.58}$ | $67.81_{\pm2.12}$ | 4.7 |
| HDS$^{ode}$ | $\mathbf{68.92_{\pm1.28}}$ | $\mathbf{83.05_{\pm0.53}}$ | $\mathbf{76.75_{\pm1.07}}$ | $\mathbf{44.26_{\pm2.11}}$ | $\mathbf{42.30_{\pm2.92}}$ | $\mathbf{93.85_{\pm0.50}}$ | $\mathbf{69.52_{\pm1.19}}$ | 1.0 |

The proof is provided in Appendix B. The system is stable when the real part of the eigenvalue is less than $0$. If the eigenvalues have additional non-zero imaginary parts, the system will oscillate and the oscillation will decrease with time. The number of $0$ eigenvalues is then determined to further investigate the characteristics of the diffusion matrix.

**Proposition 5.2.** *The multiplicity of $0$ eigenvalues after eigendecomposition of the diffusion matrix $\boldsymbol{A}$ is equal to the number of connected components in the hypergraph.*

The proof is provided in Appendix C. This property is the same as the property of our commonly used graph Laplacian matrix. Once all of the hypergraph's vertices are reachable from one other, the hypergraph has only one connected component and the multiplicity of $0$ eigenvalue in $\boldsymbol{A}$ is $1$. For ODE containing only diffusion terms as follows:

$$\begin{bmatrix} \dot{\boldsymbol{X}}_v \\ \dot{\boldsymbol{X}}_e \end{bmatrix} = \boldsymbol{A} \begin{bmatrix} \boldsymbol{X}_v(t) \\ \boldsymbol{X}_e(t) \end{bmatrix} \quad \text{with solution} \quad \begin{bmatrix} \boldsymbol{X}_v \\ \boldsymbol{X}_e \end{bmatrix} = e^{t\boldsymbol{A}} = \sum_{i=1}^{|\mathcal{V}|+|\mathcal{E}|} e^{\lambda_i t} \boldsymbol{u}_i \boldsymbol{u}_i^\top. \tag{8}$$

If $Re(\lambda_i) < 0$, there is $\lim_{t\to\infty} e^{\lambda_i t} = 0$, while $\lim_{t\to\infty} e^{\lambda_i t} = 1$ for $\lambda_i = 0$. This indicates that the representations are stable to the state corresponding to the $0$ eigenvalue by the diffusion. When each class of vertices in the hypergraph connects to vertices within the class, vertices in distinct classes are stabilized to various representations. Control terms are required to stabilize distinct categories of vertices to different representations if there are inter-hyperedges of classes. For the global including the diffusion step and the control step, we give a stability condition in the Appendix D.

## 5.2 COMPARISON WITH HYPERGRAPH NEURAL NETWORKS.

We formalize the relationship between HDS$^{ode}$ and hypergraph neural networks. Consider a situation where the control term is masked and the teleport probabilities $\alpha_v$ and $\alpha_e$ are both $1$, the vertex representations between every two layers include the relationship as $\boldsymbol{X}_v(t+2) = \boldsymbol{D}_v^{-1} \boldsymbol{H} \boldsymbol{D}_e^{-1} \boldsymbol{H}^\top \boldsymbol{X}_v(t)$, whose form is consistent with linear HGNN$^+$ layer (Gao et al., 2022) without learning parameters. Given that the teleport probabilities in HDS$^{ode}$ are susceptible to modification by the hypergraph structure and the control term finetunes the representation of diffusion, HDS$^{ode}$ has a better chance of producing an accurate representation than HGNN$^+$.

## 6 EXPERIMENTS

In this section, we conduct experiments and compare HDS$^{ode}$ to graph neural networks, graph ordinary differential equations, and hypergraph neural networks on various benchmarks.

## 6.1 SEMI-SUPERVISED VERTEX CLASSIFICATION.

Our following experiments concentrate on the semi-supervised vertex classification task, which aims to predict the labels of unlabeled vertices in a hypergraph given known partial vertex labels and all vertex features. The output layer $\hat{\boldsymbol{Y}}_v = \phi(\boldsymbol{Y}_v), \phi : \mathbb{R}^{|\mathcal{V}| \times c} \to \mathbb{R}^{|\mathcal{V}| \times o}$ acts on the final vertex representation to obtain the probability that the vertex belongs to each category, where $o$ represents the number of categories.

Table 2: Test accuracy (%) and standard deviation of semi-supervised vertex classification on a production setting with inductive and transductive predictions. "OOM", "prod.", "ind.", and "trans." denote "out of memory", "production", "inductive", "transductive", respectively. The best results are shown in bold.

| Model | | Cora-CA | DBLP-CA | News20 | IMDB4k-CA | IMDB4k-CD | DBLP4k-CC | DBLP4k-CP |
|---|---|---|---|---|---|---|---|---|
| GCN | prod. | $65.13_{\pm4.87}$ | $81.62_{\pm1.42}$ | $67.13_{\pm0.44}$ | $42.09_{\pm2.72}$ | $41.12_{\pm1.90}$ | $90.16_{\pm1.32}$ | $63.48_{\pm1.75}$ |
| | ind. | $64.48_{\pm4.57}$ | $81.54_{\pm1.71}$ | $67.33_{\pm0.65}$ | $41.40_{\pm3.52}$ | $42.05_{\pm1.84}$ | $90.80_{\pm1.32}$ | $62.81_{\pm2.82}$ |
| | trans. | $65.46_{\pm4.63}$ | $81.23_{\pm1.25}$ | $67.13_{\pm0.53}$ | $42.55_{\pm2.66}$ | $40.84_{\pm2.00}$ | $90.45_{\pm1.21}$ | $63.46_{\pm1.48}$ |
| GraphSAGE | prod. | $66.52_{\pm2.08}$ | $80.67_{\pm1.19}$ | $68.91_{\pm1.70}$ | $42.98_{\pm2.21}$ | $41.28_{\pm2.30}$ | $91.70_{\pm0.69}$ | $61.53_{\pm3.64}$ |
| | ind. | $67.38_{\pm0.56}$ | $80.91_{\pm1.40}$ | $68.51_{\pm1.36}$ | $45.04_{\pm2.97}$ | $41.58_{\pm2.42}$ | $91.58_{\pm0.56}$ | $62.11_{\pm4.89}$ |
| | trans. | $66.08_{\pm2.65}$ | $79.62_{\pm1.14}$ | $68.99_{\pm1.85}$ | $42.67_{\pm2.20}$ | $41.15_{\pm2.25}$ | $91.74_{\pm0.76}$ | $60.73_{\pm3.87}$ |
| GraphCON | prod. | $66.74_{\pm3.34}$ | $81.53_{\pm2.09}$ | OOM | $42.63_{\pm2.47}$ | $41.51_{\pm2.76}$ | OOM | $67.22_{\pm1.87}$ |
| | ind. | $64.16_{\pm3.67}$ | $81.37_{\pm2.12}$ | OOM | $42.25_{\pm2.93}$ | $41.25_{\pm2.97}$ | OOM | $67.53_{\pm2.43}$ |
| | trans. | $66.82_{\pm3.44}$ | $81.58_{\pm2.09}$ | OOM | $42.79_{\pm2.42}$ | $41.58_{\pm2.85}$ | OOM | $67.14_{\pm2.22}$ |
| HGNN | prod. | $66.72_{\pm3.08}$ | $81.95_{\pm1.14}$ | $77.09_{\pm0.60}$ | $41.70_{\pm3.02}$ | $41.23_{\pm2.76}$ | $93.76_{\pm0.83}$ | $67.25_{\pm1.87}$ |
| | ind. | $66.47_{\pm2.18}$ | $81.63_{\pm1.28}$ | $77.02_{\pm0.85}$ | $41.18_{\pm4.24}$ | $42.27_{\pm2.50}$ | $\mathbf{93.73_{\pm0.75}}$ | $65.08_{\pm3.66}$ |
| | trans. | $66.66_{\pm3.53}$ | $81.46_{\pm1.08}$ | $77.08_{\pm0.49}$ | $42.10_{\pm2.91}$ | $40.72_{\pm3.01}$ | $93.77_{\pm0.77}$ | $67.42_{\pm1.39}$ |
| HGNN$^+$ | prod. | $67.40_{\pm3.18}$ | $81.44_{\pm1.18}$ | $76.76_{\pm0.85}$ | $41.79_{\pm2.99}$ | $41.23_{\pm2.76}$ | $93.49_{\pm1.21}$ | $66.97_{\pm1.26}$ |
| | ind. | $67.05_{\pm4.12}$ | $81.33_{\pm1.18}$ | $76.56_{\pm1.13}$ | $41.87_{\pm4.51}$ | $42.27_{\pm2.50}$ | $93.38_{\pm1.09}$ | $65.79_{\pm2.76}$ |
| | trans. | $67.26_{\pm3.41}$ | $80.93_{\pm1.25}$ | $76.72_{\pm0.78}$ | $42.33_{\pm2.71}$ | $40.72_{\pm3.01}$ | $93.57_{\pm1.16}$ | $66.86_{\pm1.12}$ |
| UniGCN | prod. | $66.77_{\pm2.28}$ | $81.67_{\pm1.44}$ | $76.64_{\pm1.20}$ | $42.43_{\pm2.58}$ | $41.39_{\pm2.99}$ | $93.54_{\pm0.54}$ | $66.63_{\pm1.60}$ |
| | ind. | $66.14_{\pm4.81}$ | $81.41_{\pm1.47}$ | $76.50_{\pm1.33}$ | $43.02_{\pm2.94}$ | $41.65_{\pm3.54}$ | $93.30_{\pm1.00}$ | $66.34_{\pm2.52}$ |
| | trans. | $66.61_{\pm1.99}$ | $81.09_{\pm1.41}$ | $76.62_{\pm1.21}$ | $42.58_{\pm2.52}$ | $41.45_{\pm2.75}$ | $93.61_{\pm0.56}$ | $66.31_{\pm1.39}$ |
| UniSAGE | prod. | $68.03_{\pm2.39}$ | $82.01_{\pm1.18}$ | $76.15_{\pm1.59}$ | $\mathbf{43.14_{\pm3.19}}$ | $41.23_{\pm3.00}$ | $93.57_{\pm0.40}$ | $68.05_{\pm1.82}$ |
| | ind. | $68.21_{\pm2.85}$ | $81.97_{\pm1.45}$ | $75.97_{\pm1.26}$ | $42.16_{\pm4.47}$ | $42.01_{\pm3.58}$ | $93.30_{\pm0.48}$ | $66.92_{\pm3.83}$ |
| | trans. | $67.88_{\pm2.72}$ | $81.38_{\pm1.10}$ | $76.10_{\pm1.80}$ | $\mathbf{43.32_{\pm2.87}}$ | $40.83_{\pm2.76}$ | $93.70_{\pm0.49}$ | $68.13_{\pm1.43}$ |
| HDS$^{ode}$ | prod. | $\mathbf{69.17_{\pm3.14}}$ | $\mathbf{82.99_{\pm0.63}}$ | $\mathbf{77.16_{\pm1.04}}$ | $43.05_{\pm2.40}$ | $\mathbf{41.98_{\pm2.49}}$ | $\mathbf{93.96_{\pm0.42}}$ | $\mathbf{68.65_{\pm2.56}}$ |
| | ind. | $\mathbf{71.28_{\pm3.04}}$ | $\mathbf{83.42_{\pm0.82}}$ | $\mathbf{77.26_{\pm1.33}}$ | $\mathbf{43.71_{\pm4.22}}$ | $\mathbf{42.34_{\pm2.93}}$ | $93.26_{\pm1.01}$ | $\mathbf{67.55_{\pm2.34}}$ |
| | trans. | $\mathbf{68.29_{\pm3.81}}$ | $\mathbf{82.23_{\pm0.65}}$ | $\mathbf{77.11_{\pm1.09}}$ | $42.74_{\pm2.05}$ | $\mathbf{41.87_{\pm2.42}}$ | $\mathbf{94.10_{\pm0.73}}$ | $\mathbf{68.61_{\pm2.43}}$ |

**Datasets.** As explained below, we employ 9 publicly accessible hypergraph benchmark datasets from existing research on hypergraph neural networks, including Cora-CA and DBLP-CA from Yadati et al. (2019), News20 from Asuncion & Newman (2007), IMDB4k-CA and IMDB4k-CD from Fu et al. (2020), DBLP4k-CC and DBLP4k-CP from Sun et al. (2011), Cooking from Gao et al. (2022), and NTU from Chen et al. (2003). The details of datasets are shown in Appendix E.

**Experiment settings and details.** In the following experiments, we evaluate HDS$^{ode}$ and the compared methods in an inductive and a production setting. In both settings, we fix the total number of known label vertices in the training set and the validation set, which contains a total of $1,500$ vertices including 10 vertices per class for training. Vertices not in the training set and validation set are for test. The training, validation, and test data for each experiment are divided five times at random, and the average performance and standard deviation of each method are reported for fair comparisons. Other experiment settings, details, and implementations are in the Appendix F.

**Compared methods.** We compare HDS$^{ode}$ to a comprehensive set of baselines divided into three groups. The first group is the graph neural network (GNN) group, in which we select two popular architectures, namely Graph Convolutional Network (GCN) (Kipf & Welling, 2017) and GraphSage (Hamilton et al., 2017). In addition, we compare two ODE-based GNN models, Graph Neural Ordinary Differential Equations (GDE) (Poli et al., 2019) and Graph-Coupled Oscillator Networks (GraphCON) (Rusch et al., 2022). In the last group, we focus on methods that compute directly on the hypergraph, including Hypergraph Neural Networks (HGNN) (Feng et al., 2019), General Hypergraph Neural Networks (HGNN$^+$) (Gao et al., 2022), UniGCN (Huang & Yang, 2021), and UniSAGE Huang & Yang (2021).

### 6.1.1 VERTEX CLASSIFICATION UNDER THE TRANSDUCTIVE SETTING.

Table 1 shows the accuracy and average ranking among different methods on 7 public datasets of the transductive vertex classification task. HDS$^{ode}$ ranks first in all datasets, surpassing hypergraph

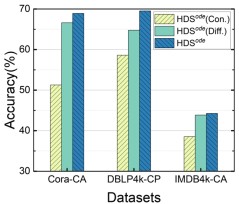 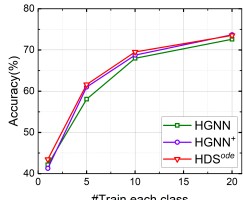 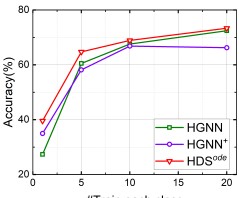 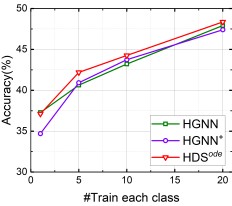

(a) On control steps and diffusion steps.

(b) On different vertex numbers of DBLP4k-CP in training set.

(c) On different vertex numbers of Cora-CA in training set.

(d) On different vertex numbers of IMDB4k-CA in training set.

Figure 3: Influence of different parts of HDS$^{ode}$.

neural network, graph ODE, and graph neural network methods. Two hypergraph neural network techniques, HGNN$^+$ and HGNN, with average rankings of 3.7 and 4.0 respectively, come in second and third place, respectively. In the following ablation experiments, we only compare HDS$^{ode}$ with these two methods. GraphCON is the graph ODE approach that ranks after HGNN$^+$ and HGNN but outperforms the two hypergraph neural network methods (UniGCN and UniSAGE), even though relational structures can only use graphs that are less expressive than hypergraphs. Among the various comparison methods, two graph neural network methods (GCN and GraphSAGE) place last, with average rankings of just 7.6 and 8.0, respectively. We first fix the correlation structure and conduct two comparisons, namely HDS$^{ode}$ vs. hypergraph neural network and graph ODE vs. graph neural network. It's important to emphasize that ODE-based methods show a clear advantage, which suggests that the continuity of neural networks strengthens vertex representations. Then we engage in another two sets of comparisons, namely HDS$^{ode}$ vs. graph ODE and hypergraph neural network vs. graph neural network. In both comparisons, the hypergraph-based methods outperform their graph-based counterparts, indicating that hypergraphs have a richer ability to represent the correlation relationship than graphs. In essence, binary correlation cannot adequately represent high-order correlation. Furthermore, even though graph ODE methods employ continuous processing in graph structures, they still fall below the best hypergraph neural network methods, demonstrating the significance of hypergraph structure.

### 6.1.2 VERTEX CLASSIFICATION UNDER THE PRODUCTION SETTING.

The transductive setting excludes predictions for unseen vertices. For a more comprehensive evaluation in real-world production scenarios, we set up a production setting experiment that includes production, transductive, and inductive predictions, where the detailed description is presented in Appendix F.2. Table 2 delineates the performance of various methods on the vertex classification task under the production setting. HDS$^{ode}$ achieves the best results across the datasets in most cases, except for production and transductive results in IMDB4k-CA, and the inductive result in DBLP4k-CC. HDS$^{ode}$ still demonstrates advantages over hypergraph neural networks and graph ODE methods in inductive prediction. In DBLP4k-CC, the inductive result of HDS$^{ode}$ is marginally below HGNN. We hypothesize this is due to the larger hyperedge in DBLP4k-CC, implying that removing inductive vertices has a greater impact on HDS$^{ode}$ diffusion process than HGNN.

### 6.2 ABLATION STUDIES

**How does the number of layers affect HDS$^{ode}$ and hypergraph neural networks?** We found that HDS$^{ode}$ has the ability to obtain long-distance neighbor knowledge. We experiment on HDS$^{ode}$ and the two best comparison methods HGNN and HGNN$^+$ in Cora-CA, with different numbers of layers. The results are shown in Figure 1. It is worth mentioning that the number of layers represents the farthest distance for each vertex to obtain neighbor information. Additionally, the number of layers in HDS$^{ode}$ also corresponds to the termination time $T$. The hypergraph neural network methods HGNN and HGNN$^+$ only receive competitive results in the shadow neural networks, which perform best at 2 layers and rapidly drop when more than 4 layers. This means that each vertex only benefits from its local neighbors and not the full hypergraph. In contrast, HDS$^{ode}$ has achieved competitive outcomes in shallow layers although not stabilized. As the number of layers increases, it acquires long-distance neighbors and brings additional benefits. Finally, when the number of layers exceeds

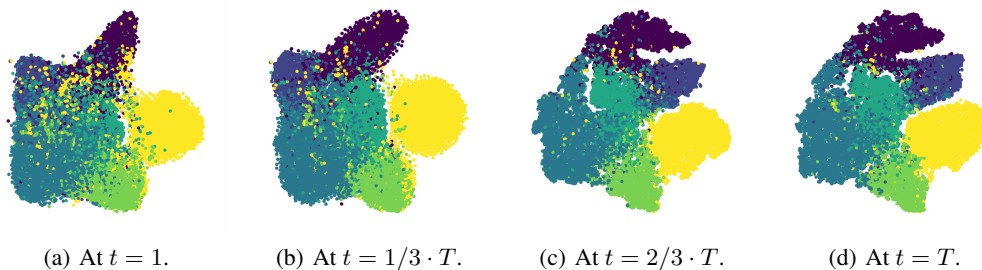

(a) At $t = 1$.     (b) At $t = 1/3 \cdot T$.     (c) At $t = 2/3 \cdot T$.     (d) At $t = T$.

Figure 4: T-SNE visualization of vertex representation at different timestamp $t$.

16, HDS$^{ode}$ produces reliable results. In this situation, each vertex collects knowledge from all of the vertex in the hypergraph.

**How are the contributions of the control steps and the diffusion steps?** We found that both the control step and the diffusion step are indispensable in HDS$^{ode}$. Figure 3(a) depicts the results of HDS$^{ode}$ using only the control steps and the diffusion steps in different benchmarks, denoted by "HDS$^{ode}$(Con.)" and "HDS$^{ode}$(Diff.)", respectively. It has been observed that removing either step reduces the results. Comparing the contributions of two steps, the result is lower if just the control step is included, since it does not employ the hypergraph structure but simply initial vertex features.

**How does the label rate affect HDS$^{ode}$?** When varying the HDS$^{ode}$ training set label rate from 1 to 20 labels per class, we observe that the performance gradually increases (see Figure 3(b),3(c),3(d)). We notice that HDS$^{ode}$ tends to improve performance more for 1, 5 or 10 labels in each class. This is because when the number of labeled vertices is relatively limited, the distance between each unlabeled vertex and the nearest labeled vertex grows. If vertices wish to profit from label information, the layer number is required to be increased, which is challenging for HGNNs. Detailed results can be found in the Appendix G.2.

### 6.3 FEATURE VISUALIZATION.

We found the vertex representation evolution of the hypergraph dynamic system based on T-SNE (Van der Maaten & Hinton, 2008) visualization. Figure 4 depicts vertex representations in DBLP-CA at various timestamps. DBLP-CA is chosen since it contains the most vertices to provide more trustworthy visualization results. It has been discovered that as time goes from 1 to $T$, various types of vertices increasingly congregate. Figure 4(a) shows points of various colors interlaced, making it difficult to distinguish between various kinds of vertices at first. After a time interval of about $T/3$, vertices of the same subclass gradually are moved closer, wherein the yellow region only contains a few vertices that are accidentally entered. When the timestamp rises close to $2T/3$ or $T$, the boundaries between the categories become more acute and the regions of different types of vertices interact with relatively few vertices from other categories. The vertex representation is accurate at the moment, and the vertex classification effectiveness is at its peak.

## 7 CONCLUSION

In this paper, we first introduce hypergraph dynamic systems, which bridge hypergraphs and dynamic systems. We then provide a specific hypergraph dynamic system based on a control-diffusion ODE. Based on this, we propose a neural implementation framework HDS$^{ode}$, which has the ability to capture long-range correlations among vertices. We introduce the properties of HDS$^{ode}$, including stability analysis and relationship with hypergraph neural networks, which are essential to understanding HDS$^{ode}$. HDS$^{ode}$ has been evaluated on vertex classification tasks in different settings. HDS$^{ode}$ has been demonstrated to obtain stable performance with increased layers while HGNNs are not controllable and stable with more layers. The evolutionary process is demonstrated by feature visualization. Future directions include extending HDS$^{ode}$ to other classes of differential equations (*e.g.*, partial differential equations) and to the time-varying setting of hyperedges.

ACKNOWLEDGMENTS

This work was supported by National Natural Science Funds of China (No. 62021002, 62088102), Beijing Natural Science Foundation (No. 4222025).

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

# A  THE ALGORITHM OF HDS$^{ode}$

---

**Algorithm 1** The algorithm of HDS$^{ode}$ framework

---

**Input:** A Hypergraph $\mathcal{G} = (\mathcal{V}, \mathcal{E})$ with an incidence matrix $\boldsymbol{H}$, a vertex feature matrix $\boldsymbol{Z}_v$, a hyperedge features matrix $\boldsymbol{Z}_e$, a degree matrix of vertices $\boldsymbol{D}_v$, and a degree matrix of hyperedges $\boldsymbol{D}_e$. The stop time of the dynamic system $T$. The teleport probabilities $\alpha_v, \alpha_e$. The control step interval $s$.

**Output:** The vertex representations $\boldsymbol{Y}_v$. The hyperedge representations $\boldsymbol{Y}_e$.

 1: Initialize vertex and hyperedge representations by $\boldsymbol{X}_v(0) \leftarrow \boldsymbol{Z}_v, \boldsymbol{X}_e(0) \leftarrow \boldsymbol{Z}_e$.
 2: **for** $t = 0$ to $T - 1$ **do**
 3:     **if** Current is the initial time or time $s$ has elapsed since the last control step **then**
 4:         The control step is performed according to equation 6.
 5:     **else**
 6:         Skip the control step by $\boldsymbol{X}_v(t + \frac{1}{2}) \leftarrow \boldsymbol{X}_v(t), \boldsymbol{X}_e(t + \frac{1}{2}) \leftarrow \boldsymbol{X}_e(t)$.
 7:     **end if**
 8:     The diffusion step is performed according to equation 7.
 9: **end for**
10: The representaions at time $T$ are the final results by $\boldsymbol{Y}_v \leftarrow \boldsymbol{X}_v(T), \boldsymbol{Y}_e \leftarrow \boldsymbol{X}_e(T)$.
11: **return** The vertex representations $\boldsymbol{Y}_v$ and the hyperedge representations $\boldsymbol{Y}_e$.

---

# B  PROOF OF PROPOSITION 5.1

*Proof.* For the diffusion matrix in HDS$^{ode}$, it is in the following form:

$$\boldsymbol{A} = \begin{bmatrix} -\alpha_v \boldsymbol{I} & \alpha_v \boldsymbol{D}_v^{-1} \boldsymbol{H} \\ \alpha_e \boldsymbol{D}_e^{-1} \boldsymbol{H}^\top & -\alpha_e \boldsymbol{I} \end{bmatrix}. \tag{9}$$

Consider the sum of all rows except the elements on the diagonal. For all $i \in \{1, 2, \ldots, |\mathcal{V}|\}$,

$$\sum_{j \neq i} |A_{i,j}| = \alpha_v, \quad \text{since} \quad d(v) = \sum_{e \in \mathcal{E}} H_{v,e}. \tag{10}$$

Similarly, For all $i \in \{|\mathcal{V}| + 1, |\mathcal{V}| + 2, \ldots, |\mathcal{V}| + |\mathcal{E}|\}$,

$$\sum_{j \neq i} |A_{i,j}| = \alpha_e, \quad \text{since} \quad \delta(e) = \sum_{v \in \mathcal{V}} H_{v,e}. \tag{11}$$

According to the Gershgorin circle theorem, the eigenvalues $\lambda_i$ of the matrix $\boldsymbol{A}$ satisfies $\lambda_i \in \mathbb{R}_\mathcal{V} \cup \mathbb{R}_\mathcal{E}$. Here

$$\mathbb{R}_\mathcal{V} = \{z \in \mathbb{C} : |z - A_{i,i}| \leq \sum_{j \neq i} |A_{i,j}| \wedge i \in \{1, 2, \ldots, |\mathcal{V}|\}\} = \{z \in \mathbb{C} : \|z + \alpha_v\|_2 \leq \alpha_v\}, \tag{12}$$

$$\mathbb{R}_\mathcal{E} = \{z \in \mathbb{C} : |z - A_{i,i}| \leq \sum_{j \neq i} |A_{i,j}| \wedge i \in \{|\mathcal{V}| + 1, \ldots, |\mathcal{V}| + |\mathcal{E}|\}\} = \{z \in \mathbb{C} : \|z + \alpha_e\|_2 \leq \alpha_e\}. \tag{13}$$

It is noticed that the two sets $\mathbb{R}_\mathcal{V}$ and $\mathbb{R}_\mathcal{E}$ have similar forms, therefore they are merged further as:

$$\lambda_i \in \mathbb{R}_\mathcal{V} \cup \mathbb{R}_\mathcal{E} = \{z \in \mathbb{C} : \|z + \max(\alpha_v, \alpha_e)\|_2 \leq \max(\alpha_v, \alpha_e)\}. \tag{14}$$

Therefore, the eigenvalue $\lambda_i$ of $\boldsymbol{A}$ is $0$ or distributed in the left half-plane of the complex plane (*i.e.*, the real part of $\lambda_i < 0$). $\qquad\square$

# C  PROOF OF PROPOSITION 5.2

*Proof.* In the proof of this proposition, we prove it from two directions, namely the multiplicity of $0$ eigenvalues is greater or equal to the connected components of the hypergraph, and the multiplicity of $0$ eigenvalues is less or equal to the connected components of the hypergraph.

We first prove the multiplicity of $0$ eigenvalues is greater or equal to the connected components of the hypergraph. Considering that the hypergraph $\mathcal{G} = (\mathcal{V}, \mathcal{E})$ contains $k$ connected components $\mathcal{G}_1 = (\mathcal{V}_1, \mathcal{E}_1), \mathcal{G}_2 = (\mathcal{V}_2, \mathcal{E}_2), \ldots, \mathcal{G}_k = (\mathcal{V}_k, \mathcal{E}_k)$, we can define $k$ eigenvectors $\boldsymbol{u}_1, \boldsymbol{u}_2, \ldots, \boldsymbol{u}_k$ as follows:

$$u_i(v) = \frac{1}{\sqrt{|\mathcal{V}_i| + |\mathcal{E}_i|}}, \forall v \in \mathcal{V}_i \quad \text{and} \quad u_i(e) = \frac{1}{\sqrt{|\mathcal{V}_i| + |\mathcal{E}_i|}}, \forall e \in \mathcal{E}_i. \tag{15}$$

According to the definition of $\boldsymbol{A}$, $\boldsymbol{A}\boldsymbol{u}_i = \boldsymbol{0}$ indicates that $\boldsymbol{u}_i$ is the corresponding eigenvector of a zero eigenvalue. According to the size of the vertices and hyperedges in the connected component $i$, it can be known that

$$||\boldsymbol{u}_i|| = \sqrt{\sum_{v \in \mathcal{V}} u_i^2(v) + \sum_{e \in \mathcal{E}} u_i^2(e)} = \sqrt{\sum_{v \in \mathcal{V}_i} \frac{1}{|\mathcal{V}_i| + |\mathcal{E}_i|} + \sum_{e \in \mathcal{E}_i} \frac{1}{|\mathcal{V}_i| + |\mathcal{E}_i|}} = 1. \tag{16}$$

Since any two connected components don't include the same vertices or hyperedges, any two eigenvectors don't contain non-zero values at the same element, that is

$$\boldsymbol{u}_i^T \boldsymbol{u}_j = 0, \forall i \neq j. \tag{17}$$

Each connected component of the hypergraph can construct an eigenvector corresponding to a $0$ eigenvalue, so the multiplicity of $0$ eigenvalues is greater or equal to the connected components of the hypergraph.

Then we prove the multiplicity of $0$ eigenvalues is less or equal to the connected components of the hypergraph. We multiply the rows of matrix $\boldsymbol{A}$ by a different coefficient to obtain the auxiliary proof matrix $\boldsymbol{A}'$ with the same $0$ eigenvalue multiplicity, as follows:

$$\boldsymbol{A}' = \begin{bmatrix} \boldsymbol{D}_v & -\boldsymbol{H} \\ -\boldsymbol{H}^\top & \boldsymbol{D}_e \end{bmatrix}. \tag{18}$$

$\boldsymbol{A}'$ is a positive semidefinite matrix because it satisfies

$$\boldsymbol{\xi}^T \boldsymbol{A}' \boldsymbol{\xi} = \sum_{i,j} \xi(i) A'_{i,j} \xi(j) = \sum_{v \in e} (\xi(v) - \xi(e))^2 \geq 0, \tag{19}$$

where the condition for the equality is that the element of the vector $\boldsymbol{\xi}$ corresponding to the vertex and hyperedge in each connected component is a constant. It can be noted that the eigenvectors $\boldsymbol{u}_1, \boldsymbol{u}_2, \ldots, \boldsymbol{u}_k$ of $\boldsymbol{A}$ are also the eigenvectors of $\boldsymbol{A}'$. Assume that the multiplicity of $0$ eigenvalues in matrix $\boldsymbol{A}'$ is greater than the number of connected components in the hypergraph, then

$$\exists \boldsymbol{\xi} \neq \boldsymbol{0}, \quad \boldsymbol{\xi}^T \boldsymbol{A}' \boldsymbol{\xi} = 0 \quad \wedge \quad \boldsymbol{\xi} \perp \boldsymbol{u}_1, \boldsymbol{u}_2, \ldots, \boldsymbol{u}_k. \tag{20}$$

As $\boldsymbol{\xi} \neq \boldsymbol{0}$, there exists $\xi(v) \neq 0$ or $\xi(e) \neq 0$. Let's assume that there are non-zero elements of the vertices or hyperedges in the $i$-th connected component in $\boldsymbol{\xi}$. Since the element of the eigenvectors $\boldsymbol{\xi}$ corresponding to the vertex and hyperedge in each connected component is a constant, $\boldsymbol{\xi}$ and $\boldsymbol{u}_i$ satisfy $\boldsymbol{\xi}^T \boldsymbol{u}_i \neq 0$, which contradicts the hypothesis. So the multiplicity of $0$ eigenvalues is less or equal to the connected components of the hypergraph.

According to the proofs in the above two directions, there is only one possibility that the multiplicity of $0$ eigenvalues of matrix $\boldsymbol{A}$ is equal to the number of connected components of the hypergraph. $\qquad \square$

## D  A STABILITY CONDITION OF THE DYNAMIC SYSTEM

We consider the overall ODE system as represented by $\dot{\boldsymbol{X}} = \boldsymbol{A}\boldsymbol{X} + g(\boldsymbol{X})$, where $\dot{\boldsymbol{X}}$ encapsulates both the vertex and hyperedge rate of change $[\dot{\boldsymbol{X}}_v, \dot{\boldsymbol{X}}_e]^\top$, and $g = \sigma(\boldsymbol{X}\boldsymbol{W} + \boldsymbol{b})$ denotes the control function modulating vertex and hyperedge representations. Here, the control function employs an activation function defined by $\sigma(\cdot) = \max(0, \cdot)$. Notably, in instances where the control term nullifies, system dynamics revert to being just diffusion-driven, thus reverting to a stability condition ruled only by the diffusion term.

For scenarios divergent from this particular case, we front a Sylvester system defined by $\dot{\boldsymbol{X}} = \boldsymbol{A}\boldsymbol{X} + \boldsymbol{X}\boldsymbol{W}$, promoting further analysis based on the prior work of Kanuri et al. (2020). We embark on solving the constituent subsystems $\dot{\boldsymbol{X}} = \boldsymbol{A}\boldsymbol{X}$ and $\dot{\boldsymbol{X}} = \boldsymbol{W}^*\boldsymbol{X}$ respectively, delineating

$Y_1(t)$ and $Y_2(t)$ as their respective solutions, with $W^*$ standing for the transpose of the complex conjugate of $W$. We introduce the notion that a matrix $Y_1$ is deemed $\Psi$-bounded on $\mathbb{R}$ if there exists a bound for $\Psi(t)Y_1(t)$ over $\mathbb{R}$. Specifically, a scalar $M > 0$ such that $\sup_{t \in \mathbb{R}} |\Psi(t)Y_1(t)| \leq M$. Similarly, a matrix function $(\Phi, \Psi)$ is characterized as $(\Phi, \Psi)$-bounded if $\|\Phi Y_1 Y_2^* \Psi^*\|$ remains bounded on $\mathbb{R}$.

The stability of the system is predicated on two key conditions. Firstly, the continuity of matrices $A$ and $W$ on $\mathbb{R}$, which comports with our definitions. Secondly, the existence of a positive scalar $K$ such that for all $t \geq 0$, the integral $\int_{-\infty}^{\infty} \|\Phi(t)Y_1(t)P Y_2^*(t)\Psi^*(t)\| \, dt$, remains bounded by $K$. Here, $\Phi(t)$ and $\Psi^*(t)$ are the bounded solutions for $Y_1(t)$ and $Y_2(t)$, respectively.

## E  DATASETS

Table S1: Real-world hypergraph benchmark dataset statistics.

| Dataset | Cora-CA | DBLP-CA | News20 | IMDB4k-CA | IMDB4k-CD | DBLP4k-CC | DBLP4k-CP | Cooking | NTU |
|---|---|---|---|---|---|---|---|---|---|
| **#Vertices** | $2,708$ | $41,302$ | $16,342$ | $4,278$ | $4,278$ | $4,057$ | $4,057$ | $7,304$ | $2,012$ |
| **#Hyperedges** | $1,072$ | $22,363$ | $100$ | $5,257$ | $2,081$ | $20$ | $14,328$ | $2,755$ | $3,345$ |
| **#Features** | $1,433$ | $1,425$ | $1,433$ | $3,066$ | $3,066$ | $334$ | $334$ | $7,304$ | $6,144$ |
| **#Classes** | $7$ | $6$ | $4$ | $3$ | $3$ | $4$ | $4$ | $20$ | $67$ |

Table S2: The train/val/test statistics of datasets in the transductive setting.

| Dataset | #Vertices | #Hyperedges | #Train/Val/Test Vertices | Split Ratio (%) |
|---|---|---|---|---|
| Cora-CA | $2,708$ | $1,072$ | $70/1,430/1,208$ | $2.6/52.8/44.6$ |
| DBLP-CA | $41,302$ | $22,363$ | $60/1,440/39,802$ | $0.1/3.5/96.4$ |
| News20 | $16,342$ | $100$ | $40/1,460/14,842$ | $0.3/8.9/90.8$ |
| IMDB4k-CA | $4,278$ | $5,257$ | $30/1,470/2,778$ | $0.7/34.4/64.9$ |
| IMDB4k-CD | $4,278$ | $2,081$ | $30/1,470/2,778$ | $0.7/34.4/64.9$ |
| DBLP4k-CC | $4,057$ | $20$ | $40/1,460/2,557$ | $1.0/36.0/63.0$ |
| DBLP4k-CP | $4,057$ | $14,328$ | $40/1,460/2,557$ | $1.0/36.0/63.0$ |

Cora-CA and DBLP-CA are obtained from Yadati et al. (2019) targeting paper classification, in which each paper is regarded as a vertex of the hypergraph, the hyperedge represents the co-author relationship of the paper, and each vertex's feature is the bag-of-words representation of the related paper. The News20 is from Asuncion & Newman (2007) with the goal being to classify each news, in which each vertex represents a news message, the hyperedge represents the occurrence of 100 popular words among news, and the vertex features are from the TF-IDF representations of the news content. The IMDB4k-CA and IMDB4k-CD datasets have been developed by Fu et al. (2020), whose aim is to categorize each film into action, comedy, and drama. Both datasets consider movies as vertices with vertex features as bag-of-words representations of movie narrative keywords, and hyperedges are conducted based on common actors and common directors, respectively. DBLP4k-CC and DBLP4k-CP are academic networks to classify authors into 4 areas, namely databases, data mining, machine learning, and information retrieval (Sun et al., 2011). The datasets treat each vertex as an author, and hyperedges are provided based on co-conferences attended and co-publications among authors in two datasets, respectively. The Cooking dataset consists of vertices representing dishes, with hyperedges indicating dishes that use the same ingredients. Each dish is also associated with categorical information indicating its cuisine type, such as French, Japanese. This dataset poses a unique challenge as it lacks initial vertex features. The NTU dataset (Chen et al., 2003) includes 3D shapes categorized into various classes like chairs, doors, etc. The vertex features are extracted using Multi-View Convolutional Neural Networks (MVCNN) (Su et al., 2015) and Group-View Convolutional Neural Networks (GVCNN) (Feng et al., 2018) for 3D shapes. Since the NTU dataset does not come with an initial hypergraph structure, we constructed it by treating each 3D shape as a vertex and using a k-nearest neighbors method to build hyperedges based on MVCNN features and

GVCNN features, respectively, thereby establishing the hypergraph structure. The detailed statistics of 9 benchmarks are in Table S1.

## F  EXPERIMENT SETTINGS AND DETAILS

### F.1  TRANSDUCTIVE SETTING

The transductive setting is one of the most common experimental settings in graph/hypergraph vertex classification. In the transductive setting, the graph is fixed at training and testing. We know the features of every node in the graph, but we only know the labels of a portion of the vertices (*i.e.*, the vertices in the training set, and the validation set). The goal is to predict the labels of the remaining unlabeled vertices in the hypergraph.

In our experiment, $1,500$ vertices are considered as vertices with known labels for each dataset, which represent the union of the training and validation sets. We select 10 vertices for each category as the training set among the known labeled vertices. The test set is composed of vertices not included in the training set or validation set. Table S2 provides the statistics of datasets in the transductive setting.

### F.2  PRODUCTION SETTING

Table S3: The train/val/test statistics of datasets in the production setting.

| Dataset | #Vertices | #Hyperedges | #Train/Val/Trans/Ind Vertices | Split Ratio (%) |
|---------|-----------|-------------|-------------------------------|-----------------|
| Cora-CA | $2,708$ | $1,072$ | $70/1,430/966/242$ | 2.6/52.8/35.7/8.9 |
| DBLP-CA | $41,302$ | $22,363$ | $60/1,440/31,842/7,960$ | 0.1/3.5/77.1/19.3 |
| News20 | $16,342$ | $100$ | $40/1,460/11,872/2,968$ | 0.3/8.9/72.6/18.2 |
| IMDB4k-CA | $4,278$ | $5,257$ | $30/1,470/2,222/556$ | 0.7/34.4/51.9/13.0 |
| IMDB4k-CD | $4,278$ | $2,081$ | $30/1,470/2,222/556$ | 0.7/34.4/51.9/13.0 |
| DBLP4k-CC | $4,057$ | $20$ | $40/1,460/2,046/511$ | 1.0/36.0/50.4/12.6 |
| DBLP4k-CP | $4,057$ | $14,328$ | $40/1,460/2,046/511$ | 1.0/36.0/50.4/12.6 |

Despite being a common research setting for vertex classification, the transductive setting excludes predictions for hidden vertices. As a result, we also take into account how well HDS$^{ode}$ is in actual production settings, including transductive and inductive forecasts.

To evaluate the model inductively, we retain a few test vertices from training to form an inductive set, and the remaining test vertices are treated as the observed transductive set (*i.e.*, $\mathcal{V}_U = \mathcal{V}_{trans} \sqcup \mathcal{V}_{ind}$ where $\mathcal{V}_U, \mathcal{V}_{trans}, \mathcal{V}_{ind}$ represent unlabeled test vertex set, transductive vertex set, and inductive vertex set, respectively). Since the model is retrained periodically in production, $\mathcal{V}_{ind}$ is regarded as the fresh unobserved set of vertices that enter the hypergraph between two pieces of training. For a hypergraph, we need to remove the inductive vertex set and its associated hyperedges to obtain the observable hypergraph $\mathcal{G}_{obs} = (\mathcal{V}_{obs}, \mathcal{E}_{obs})$.

Here, we formally present training and testing methods in production settings. For the vertices, features of the whole hypergraph dataset, we can divide it into $\mathcal{V} = \mathcal{V}_L \sqcup \mathcal{V}_{trans} \sqcup \mathcal{V}_{ind}$, $\boldsymbol{X}_v = \boldsymbol{X}_L \sqcup \boldsymbol{X}_{trans} \sqcup \boldsymbol{X}_{ind}$, where the observable vertices $\mathcal{V}_{obs} = \mathcal{V}_L \sqcup \mathcal{V}_{trans}$, the observable features $\boldsymbol{X}_{obs} = \boldsymbol{X}_L \sqcup \boldsymbol{X}_{trans}$. We use $\boldsymbol{y}_L$ to represent the labels of the vertices of the training and validation sets. To obtain transductive results, we use hypergraph $\mathcal{G}_{obs}$, feature $\boldsymbol{X}_{obs}$, and labels $\boldsymbol{y}_L$ to train the model, and then use the hypergraph $\mathcal{G}_{obs}$, and feature $\boldsymbol{X}_{obs}$ to predict the labels of the vertex set $\mathcal{V}_{trans}$. To obtain indutive results, we use hypergraph $\mathcal{G}_{obs}$, feature $\boldsymbol{X}_{obs}$, and labels $\boldsymbol{y}_L$ to train the model, and then use the hypergraph $\mathcal{G}$, and feature $\boldsymbol{X}_v$ to predict the labels of the vertex set $\mathcal{V}_{ind}$. To obtain production results, we use hypergraph $\mathcal{G}_{obs}$, feature $\boldsymbol{X}_{obs}$, and labels $\boldsymbol{y}_L$ to train the model, and then use the hypergraph $\mathcal{G}$, and feature $\boldsymbol{X}_v$ to predict the labels of the vertex set $\mathcal{V}_{trans} \sqcup \mathcal{V}_{ind}$.

In a manner akin to the transductive setting, $1,500$ vertices in the production setting embody the combined training and validation sets, with each vertex treated as having a known label within

datasets. Vertices outside the scope of the training and validation sets are designated as the test set, wherein $20\%$ serves as targets for inductive predictions. This arrangement derives from the observation that, on average, fewer vertices in inductive examination compared to transductive examination in a production setting. Table S3 provides the statistics of datasets in the production setting.

### F.3 HYPERGRAPH CLIQUE EXPANSION TO GRAPH

Given that the graph-based methods in our comparison methods are unable to execute directly on hypergraph datasets, we describe a method here that expands cliques to transform hypergraph $\mathcal{G} = (\mathcal{V}, \mathcal{E})$ into graph $\mathcal{G}_{cq} = (\mathcal{V}_{cq}, \mathcal{E}_{cq})$. The goal of a clique expansion is to transform the hyperedges in a hypergraph into graph edges. Clique expansion will add edges to any two of the hyperedge's vertices for each hyperedge. Formally, the vertex set and edge set after clique expansion can be defined as:

$$\mathcal{V}_{cq} = \mathcal{V} \quad \text{and} \quad \mathcal{E}_{cq} = \{(v_i, v_j) : \exists e \in \mathcal{E} \wedge \{v_i, v_j\} \subseteq e\}. \tag{21}$$

### F.4 TRAINING DETAILS

Table S4: The hyperparameters of HDS$^{ode}$.

| **Dataset** | Cora-CA | DBLP-CA | News20 | IMDB4k-CA | IMDB4k-CD | DBLP4k-CC | DBLP4k-CP |
|---|---|---|---|---|---|---|---|
| $\alpha_v$ | 0.05 | 0.05 | 0.1 | 0.1 | 0.05 | 0.05 | 0.1 |
| $\alpha_e$ | 0.9 | 0.9 | 0.9 | 0.9 | 0.9 | 0.9 | 0.9 |
| learning rate | $10^{-2}$ | $10^{-2}$ | $10^{-2}$ | $10^{-2}$ | $10^{-2}$ | $10^{-2}$ | $10^{-2}$ |
| weight decay | $5 \times 10^{-4}$ | $5 \times 10^{-4}$ | $5 \times 10^{-4}$ | $5 \times 10^{-4}$ | $5 \times 10^{-4}$ | $5 \times 10^{-4}$ | $5 \times 10^{-4}$ |
| dropout | 0.15 | 0.15 | 0.15 | 0.1 | 0.15 | 0.15 | 0.15 |

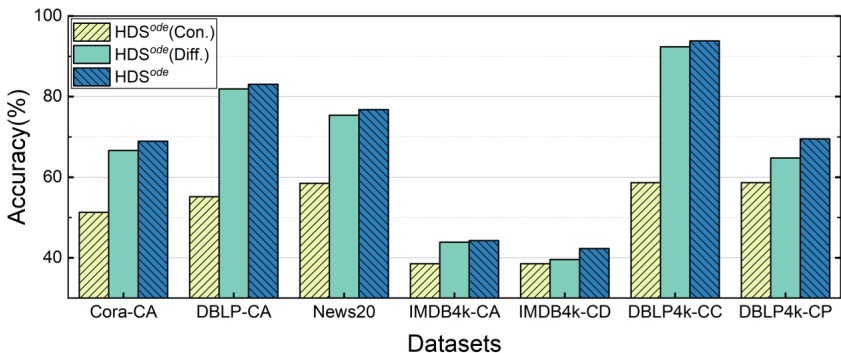

Figure S1: Comparison of the contribution of control steps and diffusion steps across all datasets.

For graph neural networks and hypergraph neural networks, the number of network layers is set to 2 to prevent over-smoothing, and the dimension of hidden layers is 64. In GDE, we set up the same network structure as in the paper with hidden layer dimension 64. In the experiment of the GraphCON, the graph convolution operator is selected as an autonomous coupling function, the number of layers is 16, the hyperparameter search range in ODE is in $\{0, 0.05, \ldots, 1\}$. In HDS$^{ode}$, the control term time interval is set to 20, the termination time $T$ is set to 40, and the search range for the hyperparameters $\alpha_v, \alpha_e$ in ODE is set to $\{0.05, 0.1, \ldots, 0.95\}$. For datasets lacking hyperedge features, the initial value of the hyperedge feature is the aggregation of features from its internal vertices (i.e., $\boldsymbol{Z}_e = \boldsymbol{D}_e^{-1} \boldsymbol{H}^\top \boldsymbol{Z}_v$). The random seed is the same for all experiments of different methods and is set to $2,022$. All models are trained for 200 epochs using Adam optimizer with learning rate in $\{10^{-2}, 10^{-3}\}$, weight decay in $\{5 \times 10^{-4}, 1 \times 10^{-4}, 5 \times 10^{-5}\}$, and dropout in $\{0.05, 0.1, \ldots 0.95\}$. The loss function is cross-entropy loss. We randomly partition the dataset five times, select hyperparameters based on the average accuracy under the validation set, and report

the average test accuracy and standard deviation under the hyperparameters. Table S4 provides the hyperparameter details of HDS$^{ode}$ in the transductive setting.

# G ADDITIONAL ABLATION STUDY RESULTS

## G.1 ABLATION STUDY ON CONTROL STEP AND DIFFUSION STEP

We further analyze the contribution of the control step and the diffusion step in 7 benchmarks, with results shown in Figure S1. In all datasets, both steps are essential. HDS$^{ode}$(Con.) results are worst since any hypergraph structure is not exploited. In IMDB4k-CA and IMDB4k-CD along with DBLP4k-CC and DBLP4k-CP, it can be noticed that HDS$^{ode}$(Con.) achieves the same effect since these two groups of datasets share the same vertex features, respectively.

## G.2 ABLATION STUDY ON THE NUMBER OF VERTICES IN THE TRAINING SET

Table S5: Test accuracy (%) and standard deviation of semi-supervised vertex classification on a transductive setting with 1 training vertex each class.

| Model | Cora-CA | DBLP-CA | News20 | IMDB4k-CA | IMDB4k-CD | DBLP4k-CC | DBLP4k-CP |
|---|---|---|---|---|---|---|---|
| GCN | $25.14_{\pm7.72}$ | $46.24_{\pm6.78}$ | $40.85_{\pm11.39}$ | $36.65_{\pm2.11}$ | $36.36_{\pm1.73}$ | $83.46_{\pm7.62}$ | $36.65_{\pm9.91}$ |
| GraphSAGE | $25.21_{\pm3.91}$ | $46.92_{\pm10.31}$ | $42.44_{\pm12.20}$ | $36.23_{\pm0.83}$ | $36.10_{\pm1.17}$ | $77.55_{\pm8.27}$ | $37.13_{\pm8.01}$ |
| HGNN | $27.35_{\pm2.53}$ | $47.88_{\pm6.49}$ | $46.63_{\pm10.01}$ | $37.28_{\pm1.97}$ | $37.12_{\pm1.73}$ | $93.32_{\pm0.85}$ | $42.15_{\pm9.86}$ |
| HGNN$^+$ | $34.98_{\pm3.93}$ | $52.73_{\pm3.96}$ | $49.50_{\pm11.17}$ | $34.70_{\pm1.84}$ | $37.31_{\pm2.33}$ | $93.37_{\pm0.35}$ | $41.92_{\pm8.02}$ |
| UniGCN | $27.94_{\pm1.31}$ | $50.91_{\pm6.86}$ | $47.08_{\pm9.65}$ | $37.02_{\pm1.70}$ | $36.83_{\pm1.47}$ | $92.73_{\pm1.15}$ | $39.83_{\pm8.70}$ |
| UniSAGE | $38.80_{\pm2.04}$ | $53.33_{\pm3.14}$ | $49.23_{\pm9.24}$ | $37.05_{\pm1.68}$ | $36.93_{\pm1.53}$ | $92.64_{\pm2.54}$ | $42.39_{\pm7.13}$ |
| HDS$^{ode}$ | $39.58_{\pm3.59}$ | $60.22_{\pm2.17}$ | $52.70_{\pm9.58}$ | $37.12_{\pm2.20}$ | $37.38_{\pm1.29}$ | $93.43_{\pm1.56}$ | $43.46_{\pm7.51}$ |

Table S6: Test accuracy (%) and standard deviation of semi-supervised vertex classification on a transductive setting with 5 training vertex each class.

| Model | Cora-CA | DBLP-CA | News20 | IMDB4k-CA | IMDB4k-CD | DBLP4k-CC | DBLP4k-CP |
|---|---|---|---|---|---|---|---|
| GCN | $54.61_{\pm4.64}$ | $73.65_{\pm6.84}$ | $62.81_{\pm2.65}$ | $39.14_{\pm2.97}$ | $38.54_{\pm2.44}$ | $89.46_{\pm1.17}$ | $56.96_{\pm5.85}$ |
| GraphSAGE | $53.11_{\pm3.74}$ | $73.58_{\pm5.40}$ | $62.66_{\pm3.73}$ | $39.72_{\pm1.16}$ | $38.51_{\pm1.34}$ | $91.08_{\pm1.03}$ | $55.95_{\pm5.33}$ |
| HGNN | $60.52_{\pm4.12}$ | $75.55_{\pm4.27}$ | $75.05_{\pm1.57}$ | $40.63_{\pm1.76}$ | $39.42_{\pm2.58}$ | $93.35_{\pm0.77}$ | $58.07_{\pm6.02}$ |
| HGNN$^+$ | $58.17_{\pm5.10}$ | $77.34_{\pm2.49}$ | $75.43_{\pm1.81}$ | $40.92_{\pm2.92}$ | $39.41_{\pm2.58}$ | $93.03_{\pm1.46}$ | $60.83_{\pm4.70}$ |
| UniGCN | $58.36_{\pm5.11}$ | $74.54_{\pm4.88}$ | $74.98_{\pm1.70}$ | $41.51_{\pm2.56}$ | $39.94_{\pm2.31}$ | $93.04_{\pm1.28}$ | $59.91_{\pm4.92}$ |
| UniSAGE | $62.56_{\pm3.27}$ | $77.94_{\pm2.39}$ | $73.90_{\pm2.85}$ | $41.67_{\pm2.50}$ | $39.56_{\pm2.85}$ | $93.28_{\pm0.42}$ | $59.74_{\pm3.22}$ |
| HDS$^{ode}$ | $64.73_{\pm2.27}$ | $78.89_{\pm2.95}$ | $75.42_{\pm2.05}$ | $42.19_{\pm1.90}$ | $41.10_{\pm2.88}$ | $93.41_{\pm0.85}$ | $61.62_{\pm1.83}$ |

Table S5 and Table S6 show the result of semi-supervised vertex classification on a transductive setting with 1 and 5 training vertex for each class, respectively. Under conditions with limited effective supervisory information, information cannot be sufficiently propagated through the network, resulting in limitations of general hypergraph neural networks. However, our ODE-based model, through the use of more layers, allows for the more extensive propagation of this limited supervisory information. This more comprehensive message passing enables our model to overcome the performance bottlenecks with the propagation capabilities of general hypergraph neural networks (with an average 1.73 and 0.82 accuracy enhancement in 1 and 5 training vertex in each class, respectively). In our experiments, we have explored scenarios with significantly fewer training samples in each class, specifically, datasets with only 5 and even 1 training vertex in each class. These few-shot conditions present a more challenging environment for learning accurate vertex representations, as the available information is considerably limited.

### G.3 ABLATION STUDY ON CHALLENGING DATASET

Table S7: Results on Cooking and NTU datasets.

| Model | Cooking | | NTU | |
|---|---|---|---|---|
| #Train Vertices | 1 | 5 | 1 | 5 |
| GCN | $25.90_{\pm6.46}$ | $32.87_{\pm3.99}$ | $61.48_{\pm4.07}$ | $79.64_{\pm1.85}$ |
| GraphSAGE | $26.68_{\pm5.91}$ | $32.73_{\pm5.11}$ | $60.83_{\pm6.53}$ | $76.78_{\pm1.13}$ |
| HGNN | $28.83_{\pm7.05}$ | $45.35_{\pm6.71}$ | $67.38_{\pm2.07}$ | $85.35_{\pm1.62}$ |
| HGNN$^+$ | $32.94_{\pm5.37}$ | $47.87_{\pm4.21}$ | $67.34_{\pm2.52}$ | $85.27_{\pm0.92}$ |
| UniGCN | $28.25_{\pm7.16}$ | $45.98_{\pm7.12}$ | $66.60_{\pm3.37}$ | $84.10_{\pm0.86}$ |
| UniSAGE | $35.26_{\pm5.00}$ | $47.26_{\pm5.19}$ | $67.96_{\pm2.96}$ | $84.26_{\pm1.80}$ |
| HDS$^{ode}$ | $\mathbf{36.15_{\pm4.18}}$ | $\mathbf{49.37_{\pm3.38}}$ | $\mathbf{71.36_{\pm1.36}}$ | $\mathbf{86.64_{\pm1.28}}$ |

We further show the performance of our model in diverse scenarios. Specifically, we have included tests on datasets with unique characteristics, without initial vertex features (Cooking dataset), and without an initial hypergraph structure (NTU dataset). The Results are shown in Table S7. In our experimental evaluation, we observe significant performance improvements in both the Cooking and NTU datasets when using our model. Specifically, when training with only 5 samples per class, our model achieves an enhancement of $3.13\%$ on the Cooking dataset and $1.51\%$ on the NTU dataset. With only 1 sample per class, the performance gains are even more pronounced, $2.52\%$ on the Cooking dataset and a $5.00\%$ on the NTU dataset. This indicates a notable enhancement in our model's ability to capture and propagate limited supervisory information effectively across the hypergraph structure, even in scenarios with minimal training data. Moreover, in a more challenging few-shot scenario, these results highlight our model's proficiency in leveraging the overall hypergraph structure to extract and utilize latent relational information, especially when dealing with limited known vertex labels. In the case of the NTU dataset, where we construct the hypergraph structure from 3D shape features using MVCNN and GVCNN, the performance improvement emphasizes our model's capacity to exploit complex relational patterns from visually extracted features. Overall, these results not only validate the effectiveness of our dynamical system-based hypergraph neural network in diverse settings but also illustrate its potential to address challenges in hypergraph learning tasks where general hypergraph models may be limited.

### G.4 ABLATION STUDY ON THE TELEPORT PROBABILITY

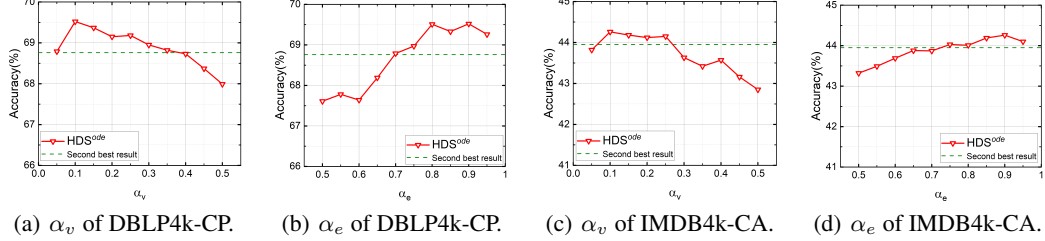

(a) $\alpha_v$ of DBLP4k-CP.  (b) $\alpha_e$ of DBLP4k-CP.  (c) $\alpha_v$ of IMDB4k-CA.  (d) $\alpha_e$ of IMDB4k-CA.

Figure S2: Additional performance comparison with teleport rates on DBLP4k-CP and IMDB4k-CA datasets.

Figure S2 shows the result of teleport probabilities $\alpha_v$ and $\alpha_e$ of HDS$^{ode}$ fall inside a certain range of $\alpha_v \in [0.05, 0.5]$ and $\alpha_e \in [0.5, 0.95]$ in DBLP4k-CP and IMDB4k-CA, respectively. It should be highlighted that HDS$^{ode}$ outperforms the second-best result across a wide range of teleport probabilities. We also notice that the teleport probability from hyperedge to vertex $\alpha_v$ is optimal when small, and conversely, the teleport probability from the vertex to the hyperedge $\alpha_e$ achieves the best results when large. This is because the initial hyperedge features are missing in DBLP4k-CP and IMDB4k-CA, and are calculated using the corresponding vertex features, which is slightly incorrect. To get the precise representations, it is evident to lower the transfer probability from inaccurate

hyperedge representation to vertices while increasing the transfer probability from accurate vertex representation to hyperedges. In general, as the hyperparameters increase, performance rises and subsequently drops. When using HDS$^{ode}$ on a fresh dataset, we can alter $\alpha_v$ to about 0.1 and $\alpha_e$ to approximately 0.9 by summarizing the results of IMDB4k-CA and DBLP4k-CP.

## G.5 ABLATION STUDY ON THE INDUCTIVE RATIO OF PRODUCTION SETTING

Table S8: The ablation study on inductive vertex ratio of production setting. We report test accuracy (%) and standard deviation of semi-supervised vertex classification on a production setting with inductive and transductive predictions. "prod.", "ind.", and "trans." denote "production", "inductive", "transductive", respectively. The best results are shown in bold.

| Inductive vertex ratio | | 0.1 | 0.2 | 0.3 | 0.4 | 0.5 | 0.6 | 0.7 | 0.8 | 0.9 |
|---|---|---|---|---|---|---|---|---|---|---|
| HGNN | prod. | $67.31_{\pm3.40}$ | $66.72_{\pm3.08}$ | $66.65_{\pm2.97}$ | $67.00_{\pm2.73}$ | $66.95_{\pm3.19}$ | $67.21_{\pm2.54}$ | $67.71_{\pm2.69}$ | $68.21_{\pm2.59}$ | $67.26_{\pm1.98}$ |
| | ind. | $65.83_{\pm4.24}$ | $66.47_{\pm2.18}$ | $66.79_{\pm2.59}$ | $66.66_{\pm3.66}$ | $66.65_{\pm3.98}$ | $66.98_{\pm3.41}$ | $67.64_{\pm3.37}$ | $68.03_{\pm3.65}$ | $67.37_{\pm2.45}$ |
| | trans. | $67.50_{\pm3.25}$ | $66.66_{\pm3.53}$ | $66.14_{\pm3.31}$ | $66.75_{\pm1.71}$ | $66.72_{\pm1.83}$ | $66.36_{\pm1.54}$ | $66.00_{\pm1.29}$ | $66.94_{\pm2.33}$ | $64.95_{\pm2.89}$ |
| HGNN$^+$ | prod. | $67.50_{\pm3.05}$ | $67.40_{\pm3.18}$ | $67.18_{\pm3.37}$ | $67.50_{\pm3.77}$ | $67.28_{\pm3.63}$ | $66.63_{\pm3.86}$ | $67.28_{\pm3.53}$ | $66.80_{\pm3.61}$ | $66.93_{\pm3.54}$ |
| | ind. | $67.00_{\pm4.98}$ | $67.05_{\pm4.12}$ | $66.51_{\pm3.47}$ | $67.03_{\pm4.06}$ | $66.75_{\pm4.01}$ | $65.93_{\pm4.56}$ | $67.24_{\pm4.01}$ | $66.52_{\pm4.40}$ | $66.88_{\pm3.97}$ |
| | trans. | $67.64_{\pm3.61}$ | $67.26_{\pm3.41}$ | $66.54_{\pm3.59}$ | $66.29_{\pm3.67}$ | $66.12_{\pm3.16}$ | $65.86_{\pm2.78}$ | $65.39_{\pm1.76}$ | $65.95_{\pm2.43}$ | $65.62_{\pm2.13}$ |
| HDS$^{ode}$ | prod. | $\mathbf{70.49_{\pm1.43}}$ | $\mathbf{69.17_{\pm3.14}}$ | $\mathbf{70.46_{\pm2.43}}$ | $\mathbf{69.33_{\pm1.97}}$ | $\mathbf{70.09_{\pm2.02}}$ | $\mathbf{69.66_{\pm3.18}}$ | $\mathbf{69.63_{\pm1.62}}$ | $\mathbf{68.49_{\pm2.40}}$ | $\mathbf{68.90_{\pm1.90}}$ |
| | ind. | $\mathbf{72.66_{\pm4.09}}$ | $\mathbf{71.28_{\pm3.04}}$ | $\mathbf{71.82_{\pm1.85}}$ | $\mathbf{69.85_{\pm1.51}}$ | $\mathbf{70.19_{\pm1.94}}$ | $\mathbf{70.52_{\pm3.00}}$ | $\mathbf{70.05_{\pm1.23}}$ | $\mathbf{68.82_{\pm2.31}}$ | $\mathbf{68.68_{\pm1.71}}$ |
| | trans. | $\mathbf{69.89_{\pm1.99}}$ | $\mathbf{68.29_{\pm3.81}}$ | $\mathbf{69.26_{\pm3.22}}$ | $\mathbf{68.52_{\pm2.88}}$ | $\mathbf{68.67_{\pm2.89}}$ | $\mathbf{67.19_{\pm3.51}}$ | $\mathbf{66.99_{\pm3.94}}$ | $\mathbf{65.20_{\pm5.08}}$ | $\mathbf{67.10_{\pm4.41}}$ |

In order to verify the impact of different inductive vertex ratios on the results under production settings, we conduct an ablation study of it on dataset Cora-CA with HGNN, HGNN$^+$, and HDS$^{ode}$. Detailed results are shown in Table S8. The table shows that, in various inductive vertex ratio settings, our HDS$^{ode}$ method outperforms both HGNN and HGNN$^+$ among production, inductive, and transductive results. Additionally, as the ratio rises, the number of the observation graph vertices $|\mathcal{V}_{obs}|$ gradually decreases, causing the three results to essentially exhibit a downward trend.

## G.6 ABLATION STUDY ON HYPERGRAPH OPERATORS AND GRAPH OPERATORS

Table S9: Test accuracy (%) and standard deviation of semi-supervised vertex classification of the ablation study on hypergraph operators and graph operators. "OOM" and "HDS$^{ode}$(graph op.)" represent "out of memory" and "HDS$^{ode}$ using graph operators", respectively. The best results are shown in bold.

| Model | Cora-CA | DBLP-CA | News20 | IMDB4k-CA | IMDB4k-CD | DBLP4k-CC | DBLP4k-CP |
|---|---|---|---|---|---|---|---|
| GCN | $65.99_{\pm3.69}$ | $82.22_{\pm1.05}$ | $67.57_{\pm0.70}$ | $43.47_{\pm2.39}$ | $41.02_{\pm2.22}$ | $90.18_{\pm1.22}$ | $64.47_{\pm0.90}$ |
| HGNN | $67.58_{\pm1.83}$ | $82.83_{\pm1.09}$ | $76.58_{\pm0.94}$ | $43.21_{\pm2.39}$ | $41.08_{\pm2.43}$ | $93.46_{\pm0.77}$ | $67.99_{\pm2.12}$ |
| HDS$^{ode}$(graph op.) | $67.48_{\pm3.04}$ | $82.78_{\pm1.93}$ | OOM | $43.81_{\pm3.16}$ | $41.53_{\pm2.18}$ | $90.67_{\pm0.83}$ | $66.75_{\pm0.79}$ |
| HDS$^{ode}$ | $\mathbf{68.92_{\pm1.28}}$ | $\mathbf{83.05_{\pm0.53}}$ | $\mathbf{76.75_{\pm1.07}}$ | $\mathbf{44.26_{\pm2.11}}$ | $\mathbf{42.30_{\pm2.92}}$ | $\mathbf{93.85_{\pm0.50}}$ | $\mathbf{69.52_{\pm1.19}}$ |

To verify the necessity of using hypergraph operators, we present an ablation study on whether the model uses hypergraph operators or graph operators in different hypergraph datasets. The ablation study includes four methods, namely GCN, HGNN, HDS$^{ode}$ using graph operators, and HDS$^{ode}$.

According to the results in Table S9., we found that in the hypergraph dataset, the hypergraph method achieves better results than the corresponding graph method (*i.e.*, HGNN beats GCN, and HDS$^{ode}$ beats HDS$^{ode}$(graph op.), respectively), which indicates that it is necessary to use hypergraph operators in hypergraph correlation structures.

As a generalization of graphs, hypergraphs have richer correlation expression capabilities than graphs. Such as in film background, hypergraphs adeptly capture the multifaceted connections among actors. Each actor can be represented as a vertex, and a hyperedge encompasses various correlations, such as actors co-starring in the same movie or originating from the same country. This hypergraph approach allows for a comprehensive representation of connections beyond pairwise interactions. Traditional graph models fall short since edges link only two nodes. They struggle to directly represent these group-based relationships without becoming complex or losing clarity. In

the graph operator, each vertex interacts with its adjacent vertices based on the edge connecting two vertices, while the hypergraph operator generates pair-wise interaction or group interaction based on the hyperedge that can connect two or more vertices.

# H DETAILED VISUALIZATION OF REPRESENTATION

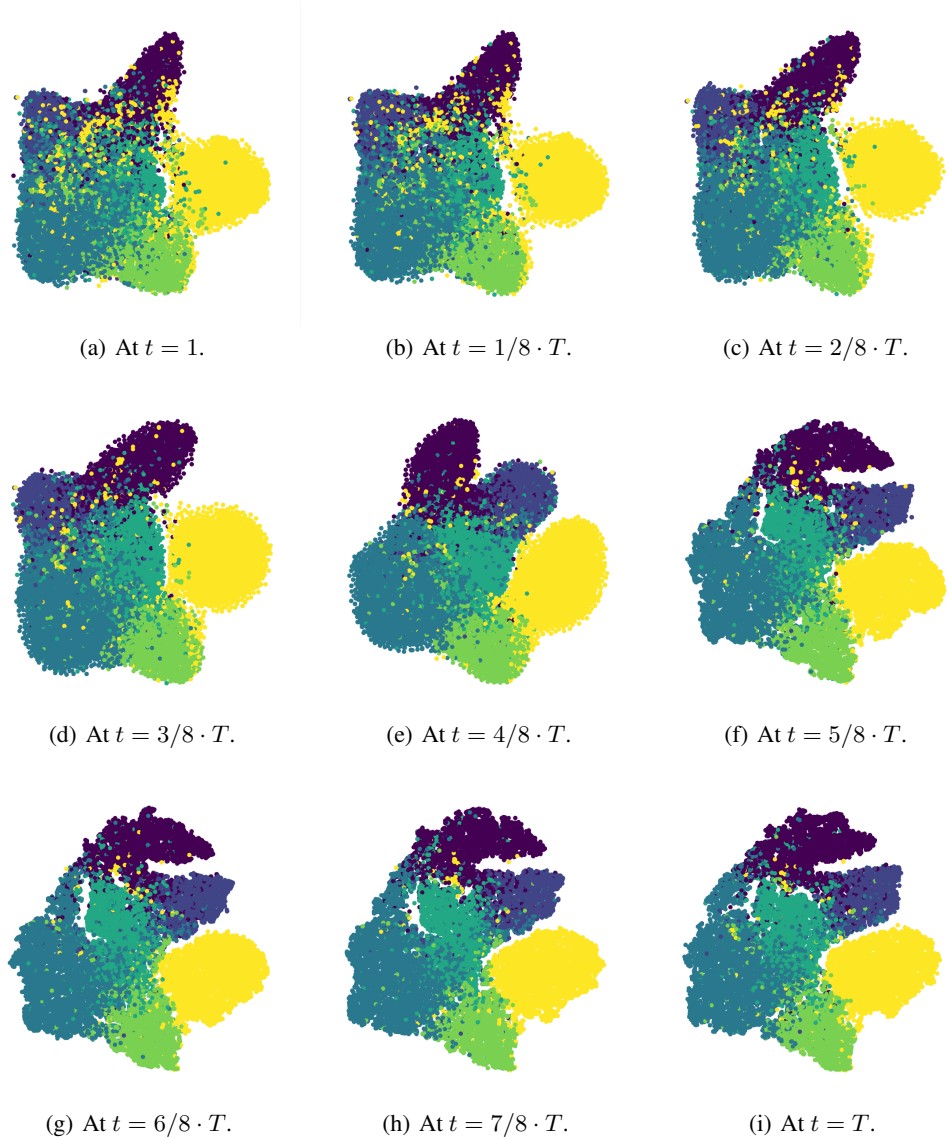

(a) At $t = 1$.      (b) At $t = 1/8 \cdot T$.      (c) At $t = 2/8 \cdot T$.

(d) At $t = 3/8 \cdot T$.      (e) At $t = 4/8 \cdot T$.      (f) At $t = 5/8 \cdot T$.

(g) At $t = 6/8 \cdot T$.      (h) At $t = 7/8 \cdot T$.      (i) At $t = T$.

Figure S3: More specific T-SNE visualization of vertex representation at different timestamp $t$.

Taking $T/8$ as the time interval, Figure S3 gives a more detailed representation evolution process in DBLP-CA. The dynamic variations in the vertex representation distribution are more clearly depicted as the time period is shortened. When the timestamp $t$ is larger, the representation is closer to stability, so the change in the same time interval is less obvious. By reducing the visualization time interval, we get more accurate evolution rules of representations. Through the accurate dynamic evolution, we provide circumstantial evidence for the evolution process of representation from raw to final.

