# OpenReview forum: "Hypergraph Dynamic System"
_ICLR.cc/2024/Conference — ICLR 2024 poster_

### Official Review · Reviewer_HBiR · 2023-10-19

**Soundness:** 3 good
**Presentation:** 3 good
**Contribution:** 3 good
**Rating:** 5
**Confidence:** 3

**Summary:**

This paper extends hypergraph neural networks (HGNNs) to model hypergraph dynamic systems. Specifically, the authors integrate the graph propagation scheme of HGNN into a control-diffusion ODE form to capture dynamics. Theoretical analysis highlights the controllability and stabilization properties of the proposed HDS$^{ode}$, which allows it to capture long-range correlations among vertices. Experimental results demonstrate the effectiveness of HDS$^{ode}$.

**Strengths:**

+ The combination of Neural ODEs with hypergraphs is an interesting idea, bringing together two distinct approaches to modeling dynamic systems.
+ The authors introduce a Lie-Trotter splitting method as the ODE solver, which is a notable contribution.
+ The theoretical analysis on stability and eigenvalue properties of hypergraphs is solid and persuasive.

**Weaknesses:**

- The proposed model appears to be a straightforward combination of existing efforts on HGNN and neural graph ODEs, essentially replacing the message-passing scheme of GNNs with a diffusive hypergraph adjacency. It would be beneficial to clarify how this combination advances the field beyond existing approaches.
- The application of hypergraph ODEs is not sufficiently motivated. Although the proposed model claims to capture system dynamics, it is evaluated on node classification tasks using static graphs, leaving the potential benefits on dynamical systems unclear.
- The experimental results show limited improvement over baseline methods, and from the comprehensive comparison it seems that variations in model structures have little influence on performance.

**Questions:**

1. What is the rationale for applying ODEs and continuous methods to static graphs? How does the model benefit from the system dynamics brought by its structure?
2. Could you provide insights into the model's performance when the hypergraph convolution is replaced with simpler graph operators like GCN, as part of ablation studies? The functionality of hypergraph convolution in the model is unclear.

---

> ### Author Response · Authors · 2023-11-17
> **Response to Reviewer HBiR (Part 1/3)**
>
> We appreciate the reviewer for providing valuable comments and feedback. We hope that our response can address your concerns.
>
> $\newline$
>
> **Response to Weakness 1**
>
> Thank you for your valuable feedback. I appreciate the opportunity to clarify how our proposed model represents a significant benefit beyond the existing HGNN and graph ODEs.
>
>
>
> **Leveraging Strengths of Hypergraph Neural Networks and Graph ODEs**. Our model leverages the strengths of both hypergraph neural networks and graph ODE. Our model captures high-order correlation beyond pairwise interactions that are often overlooked in graph models. Unlike discrete models, such as HGNN, which can only capture representation at a few hidden layers, our ODE-based hypergraph method continuously tracks the evolution of these representations until finally stable.
>
>
>
> **Effective Operation at Higher Layer Depths**. Another significant advantage of our model lies in its capacity to operate effectively at higher layer depths. This capability contrasts with most traditional (hyper)graph neural networks, which are typically constrained to lower layer depths due to issues like over-smoothing. In (hyper)graph neural networks, the restriction to fewer layers means that each vertex primarily interacts with its immediate, local neighbors. This localized view often neglects the potential influence of more distant vertices, leading to a limited understanding of the overall (hyper)graph. Our model overcomes this limitation with its capacity for higher layer depths enabled by the diffusive hypergraph.
>
>
>
> **Stability with Increased Layers**. The key advantage of our model is its stability at increased layer depths. While additional layers in most existing (hyper)graph approaches can lead to instability, our model maintains stable performance as the number of layers grows, which means that our model is more robust. This stability is crucial for delivering consistent and reliable results, marking a notable improvement over current methods and a meaningful contribution to the hypergraph field.
>
> $\newline$
>
> **Response to Weakness 2**
>
> Thank you for your insightful feedback regarding the motivation and application of hypergraph ODEs in our work. We are confident that our hypergraph ODE has the potential to advance continuous industrial scenarios involving hypergraphs. While our current evaluation focuses on static graph node classification tasks, we believe this serves as a foundational step in showcasing the stability and accuracy of our model.
>
>
>
> In industrial applications, particularly those involving hypergraphs, stability in model predictions is as crucial as accuracy. Our model's ability to provide consistent and reliable predictions across diverse data types and conditions is a key advantage. Our model is available for various tasks. The choice of vertex classification tasks for our experiment is driven by their effectiveness in evaluating the stability and performance of hypergraph neural networks across common benchmarks.
>
>
>
> In our article, we take a step towards achieving dynamic continuity in previous hypergraph neural networks by supporting the dynamic continuous representation of vertex and hyperedge representations through the ODE. We thank the reviewer for your suggestion that both hypergraph structure and representation be dynamic, which will serve as a future direction for our research.
>
> $\newline$
>
> **Response to Weakness 3**
>
> Thank you for your observations regarding the experimental results. We understand your concerns about the performance improvement over baseline methods. However, we wish to emphasize that the **primary contribution** of our work lies in achieving stable representation learning in (hyper)graphs, a general challenge in (hyper)graph neural networks [1,2].
>
>
>
> **We emphasize the dual aspects of performance** in our work: not just the direct performance metrics but also, and importantly, the stability of performance. In practical applications of hypergraph neural networks, achieving optimal performance often depends on specific conditions, a requirement that many existing methods struggle to meet consistently. This is why the robustness becomes valuable. Our focus on **robustness** and reduced dependence on such conditions ensures reliable and consistent performance across a variety of scenarios and layer depths. Further, our model not only solves the stability problem of increasing the number of layers, but also provides performance enhancement.
>
>
>
> Our model exhibits consistently high performance across an increasing number of layers, a notable advancement compared to many existing hypergraph neural networks where performance typically degrades with layer depth. This focus on maintaining stability at greater layer depths is a key advancement compared to models that only emphasize peak performance under specific conditions.

---

> ### Author Response · Authors · 2023-11-17
> **Response to Reviewer HBiR (Part 2/3)**
>
> **Response to Question 1**
>
> Thank you for your insightful query. There are several key rationales for applying ODEs and continuous methods to static hypergraphs.
>
>
>
> **Continuous Representations**. By modeling the continuous changes in vertex and hyperedge representations over time, we enable the neural network's hidden layer representation to transform continuously [3]. This approach facilitates a more effective capture of the evolving nature of the representation, ensuring stable outcomes as the number of layers increases.
>
>
>
> **Extended Interaction Range**. Furthermore, integrating ODEs and continuous methods addresses a critical limitation of general (hyper)graph neural networks, which often focus only on neighbors with low distances due to layer depth.  In general neural differential equations, the discretization method corresponds to a multi-step model [4]. Therefore, the model we construct based on ODEs allows for interactions across extended distances. This capability enables vertices to interact with long-distance neighbors, thereby broadening the interaction scope from merely local to encompassing global distances.
>
>
>
> **Fine-Tuning Through Control Term**. A distinctive distinction of our model lies in the inclusion of a control term, designed for fine-tuning representations during the diffusion process. The control term is an auxiliary function, allowing for adjustments to the diffusion process, to suit the specific dataset. This added layer of adaptability not only enhances the model's performance but also provides flexibility and precision in hypergraph representation learning.
>
>
>
> Our model integrates ODEs within the hypergraph framework. This integration results in a more nuanced representation of hidden layers, providing a smoother transition between them. This contrasts with general (hyper)graph neural networks, where features can exhibit more substantial variations between layers due to fewer intermediary layers. Second, the continuous diffusion results become stable as the number of layers in the model increases. It shows that, given sufficient depth, the model can reach an equilibrium state in system dynamics, providing consistent and reliable output.

---

> ### Author Response · Authors · 2023-11-17
> **Response to Reviewer HBiR (Part 3/3)**
>
> **Response to Question 2**
>
> We add an ablation study on whether the model uses hypergraph operators or graph operators in different hypergraph datasets. The ablation study include four methods, namely Graph Convolutional Network (GCN), Hypergraph Neural Networks (HGNN), HDS$^{ode}$ using graph operators, and HDS$^{ode}$. In the following table, "OOM" and "HDS$^{ode}$(graph op.)" represent "out of memory" and "HDS$^{ode}$ using graph operators", respectively.
>
> $\newline$
>
>
> | Model                  | Cora-CA             | DBLP-CA             | News20              | IMDB4k-CA           | IMDB4k-CD           | DBLP4k-CC           | DBLP4k-CP           |
> | ---------------------- | ------------------- | ------------------- | ------------------- | ------------------- | ------------------- | ------------------- | ------------------- |
> | GCN                    | $65.99_{\pm3.69}$   | $82.22_{\pm1.05}$   | $67.57_{\pm0.70}$   | $43.47_{\pm2.39}$   | $41.02_{\pm2.22}$   | $90.18_{\pm1.22}$   | $64.47_{\pm0.90}$   |
> | HGNN                   | $67.58_{\pm1.83}$   | $82.83_{\pm1.09}$   | $76.58_{\pm0.94}$   | $43.21_{\pm2.39}$   | $41.08_{\pm2.43}$   | $93.46_{\pm0.77}$   | $67.99_{\pm2.12}$   |
> | HDS$^{ode}$(graph op.) | $67.48_{\pm3.04}$   | $82.78_{\pm1.93}$   | OOM                 | $43.81_{\pm3.16}$   | $41.53_{\pm2.18}$   | $90.67_{\pm 0.83}$  | $66.75_{\pm0.79}$   |
> | HDS$^{ode}$            | ${68.92_{\pm1.28}}$ | ${83.05_{\pm0.53}}$ | ${76.75_{\pm1.07}}$ | ${44.26_{\pm2.11}}$ | ${42.30_{\pm2.92}}$ | ${93.85_{\pm0.50}}$ | ${69.52_{\pm1.19}}$ |
>
> $\newline$
>
> According to the results in the above table, we found that in the hypergraph dataset, the hypergraph method achieves better results than the corresponding graph method (i.e., HGNN beats GCN, and HDS$^{ode}$ beats HDS$^{ode}$(graph op.), respectively), which indicates that it is necessary to use hypergraph operators in hypergraph correlation structures.
>
>
>
> As a generalization of graphs, hypergraphs have richer correlation expression capabilities than graphs. Such as in film background, hypergraphs adeptly capture the multifaceted connections among actors. Each actor can be represented as a vertex, and a hyperedge encompasses various correlations, such as actors co-starring in the same movie or originating from the same country. This hypergraph approach allows for a comprehensive representation of connections beyond pairwise interactions. Traditional graph models fall short since edges link only two nodes. They struggle to directly represent these group-based relationships without becoming complex or losing clarity.
>
>
>
> In the graph operator, each vertex interacts with its adjacent vertices based on the edge connecting two vertices, while the hypergraph operator generates pair-wise interaction or group interaction based on the hyperedge that can connect two or more vertices. This is a key advantage of hypergraph operators.
>
> $\newline$
>
> Thank you once again for your valuable time and consideration. We hope our response addresses your concerns. We also welcome any new questions you may have.
>
> $\newline$
>
> [1] Qimai Li, Zhichao Han, Xiao-Ming Wu. "Deeper insights into graph convolutional networks for semi-supervised learning." *Proceedings of the AAAI conference on artificial intelligence*, 2018.
>
> [2] Deli Chen, Yankai Lin, Wei Li, Peng Li, Jie Zhou, Xu Sun. "Measuring and relieving the over-smoothing problem for graph neural networks from the topological view." *Proceedings of the AAAI conference on artificial intelligence*, 2020.
>
> [3] Ricky T. Q. Chen, Yulia Rubanova, Jesse Bettencourt, David K. Duvenaud. "Neural ordinary differential equations." *Advances in neural information processing systems*, 2018.
>
> [4] Yiping Lu, Aoxiao Zhong, Quanzheng Li, Bin Dong. "Beyond finite layer neural networks: Bridging deep architectures and numerical differential equations." *International Conference on Machine Learning*, 2018.

---

> ### Author Response · Authors · 2023-11-22
> **Response to Reviewer HBiR**
>
> We understand your concerns regarding the performance improvement in our experiments, and we add an additional ablation study that we have conducted to further validate our model's performance. In difficult scenarios, the hypergraph structure can better mine the complex relationships behind the data and better solve problems.
>
>
>
> **Enhanced Performance in Few-Shot Scenarios**. Under conditions with limited effective supervisory information, such information cannot be sufficiently propagated through the network, resulting in limitations of general hypergraph neural networks. However, our ODE-based model, through the use of more layers, allows for the more extensive propagation of this limited supervisory information. This more comprehensive message passing enables our model to overcome the performance bottlenecks with the propagation capabilities of general hypergraph neural networks (with average 0.95 and 1.90 accuracy enhancement in 5 and 1 training vertex each class, respectively). In our experiments, we have explored scenarios with significantly fewer training samples each class, specifically, datasets with only 5 and even 1 training vertex each class. These few-shot conditions present a more challenging environment for learning accurate vertex representations, as the available information is considerably limited. The results of these experiments are presented in the following table.
>
>
>
> **Advantages of Hypergraph Methods over Graph Methods**. In our analysis, we also observed that hypergraph methods exhibit a more significant improvement over traditional graph methods under conditions of limited effective supervisory information. This advantage stems from the inherent strengths of hypergraph modeling compared to graph modeling. Hypergraphs **transcend the pairwise modeling** capabilities of graphs, enabling a more nuanced and comprehensive representation of complex relationships within the data. The **multi-modal property** of hypergraphs allows for the integration of diverse types of interactions and dependencies, which is particularly beneficial in scenarios where conventional graph models might struggle due to their limited expressive power. These strengths of hypergraph modeling become even more pronounced in scenarios where supervisory information is sparse.
>
>
>
> **Utilizing Broader Interaction in Hypergraphs**. Our model shows a more obvious advantage over general hypergraph neural network methods. This improvement becomes clearer in these few-shot scenarios. The reason for this can be attributed to the fact that with fewer training vertices, the need to capture information beyond 1-2 hops becomes more critical since fewer labels within the neighbor range with low distance. Our model's ability to leverage a broader scope of interaction allows it to access and utilize more comprehensive information, which is especially beneficial in situations where known vertex information is scarce.
>
>
>
> Our model shows greater potential for hypergraph inference tasks with limited known vertex information, indicating its capacity to achieve superior performance in more challenging and realistic scenarios. Our model's broader interaction scope is instrumental in capturing more complex and global relationships within the data, thereby enhancing performance.
>
>
>
> In addition to the previously mentioned experiments, we show the performance of our model in diverse scenarios. Specifically, we have included tests on datasets with unique characteristics, without initial vertex features (Cooking Dataset), and without an initial hypergraph structure (NTU Dataset [1]).  Our experiments on these datasets demonstrate that our model outperforms standard hypergraph neural network approaches even in scenarios lacking initial vertex features or hypergraph structures. These results further validate the capability of our model to adapt in various complex scenarios.

---

> ### Author Response · Authors · 2023-11-22
> **Response to Reviewer HBiR**
>
> Results on 10 training vertex each class (reported in our paper).
>
> |    Model    |       Cora-CA       |       DBLP-CA       |       News20        |      IMDB4k-CA      |      IMDB4k-CD      |      DBLP4k-CC      |      DBLP4k-CP      |
> | :---------: | :-----------------: | :-----------------: | :-----------------: | :-----------------: | :-----------------: | :-----------------: | :-----------------: |
> |     GCN     |  $65.99_{\pm3.69}$  |  $82.22_{\pm1.05}$  |  $67.57_{\pm0.70}$  |  $43.47_{\pm2.39}$  |  $41.02_{\pm2.22}$  |  $90.18_{\pm1.22}$  |  $64.47_{\pm0.90}$  |
> |  GraphSAGE  |  $66.44_{\pm2.82}$  |  $81.07_{\pm1.50}$  |  $69.59_{\pm0.89}$  |  $42.05_{\pm1.95}$  |  $41.07_{\pm2.11}$  |  $92.18_{\pm0.38}$  |  $64.34_{\pm1.58}$  |
> |    HGNN     |  $67.58_{\pm1.83}$  |  $82.83_{\pm1.09}$  |  $76.58_{\pm0.94}$  |  $43.21_{\pm2.39}$  |  $41.08_{\pm2.43}$  |  $93.46_{\pm0.77}$  |  $67.99_{\pm2.12}$  |
> |  HGNN$^+$   |  $66.85_{\pm2.24}$  |  $82.40_{\pm1.27}$  |  $76.49_{\pm1.30}$  |  $43.74_{\pm1.42}$  |  $41.49_{\pm2.54}$  |  $93.46_{\pm1.09}$  |  $68.76_{\pm2.73}$  |
> |   UniGCN    |  $66.47_{\pm2.04}$  |  $82.36_{\pm1.09}$  |  $76.56_{\pm1.21}$  |  $43.34_{\pm3.26}$  |  $41.33_{\pm2.50}$  |  $93.28_{\pm0.87}$  |  $67.68_{\pm1.90}$  |
> |   UniSAGE   |  $68.59_{\pm1.61}$  |  $82.16_{\pm1.25}$  |  $75.52_{\pm1.22}$  |  $42.82_{\pm2.66}$  |  $41.62_{\pm3.05}$  |  $93.64_{\pm0.58}$  |  $67.81_{\pm2.12}$  |
> | HDS$^{ode}$ | ${68.92_{\pm1.28}}$ | ${83.05_{\pm0.53}}$ | ${76.75_{\pm1.07}}$ | ${44.26_{\pm2.11}}$ | ${42.30_{\pm2.92}}$ | ${93.85_{\pm0.50}}$ | ${69.52_{\pm1.19}}$ |
>
>
>
> Results on 5 training vertex each class.
>
> |    Model    |           Cora-CA |           DBLP-CA |            News20 | IMDB4k-CA         | IMDB4k-CD         | DBLP4k-CC         | DBLP4k-CP         | Cooking           | NTU               |
> | :---------: | ----------------: | ----------------: | ----------------: | ----------------- | ----------------- | ----------------- | ----------------- | ----------------- | ----------------- |
> |     GCN     | $54.61_{\pm4.64}$ | $73.65_{\pm6.84}$ | $62.81_{\pm2.65}$ | $39.14_{\pm2.97}$ | $38.54_{\pm2.44}$ | $89.46_{\pm1.17}$ | $56.96_{\pm5.85}$ | $32.87_{\pm3.99}$ | $79.64_{\pm1.85}$ |
> |  GraphSAGE  | $53.11_{\pm3.74}$ | $73.58_{\pm5.40}$ | $62.66_{\pm3.73}$ | $39.72_{\pm1.16}$ | $38.51_{\pm1.34}$ | $91.08_{\pm1.03}$ | $55.95_{\pm5.33}$ | $32.73_{\pm5.11}$ | $76.78_{\pm1.13}$ |
> |    HGNN     | $60.52_{\pm4.12}$ | $75.55_{\pm4.27}$ | $75.05_{\pm1.57}$ | $40.63_{\pm1.76}$ | $39.42_{\pm2.58}$ | $93.35_{\pm0.77}$ | $58.07_{\pm6.02}$ | $45.35_{\pm6.71}$ | $85.35_{\pm1.62}$ |
> |  HGNN$^+$   | $58.17_{\pm5.10}$ | $77.34_{\pm2.49}$ | $75.43_{\pm1.81}$ | $40.92_{\pm2.92}$ | $39.41_{\pm2.58}$ | $93.03_{\pm1.46}$ | $60.83_{\pm4.70}$ | $47.87_{\pm4.21}$ | $85.27_{\pm0.92}$ |
> |   UniGCN    | $58.36_{\pm5.11}$ | $74.54_{\pm4.88}$ | $74.98_{\pm1.70}$ | $41.51_{\pm2.56}$ | $39.94_{\pm2.31}$ | $93.04_{\pm1.28}$ | $59.91_{\pm4.92}$ | $45.98_{\pm7.12}$ | $84.10_{\pm0.86}$ |
> |   UniSAGE   | $62.56_{\pm3.27}$ | $77.94_{\pm2.39}$ | $73.90_{\pm2.85}$ | $41.67_{\pm2.50}$ | $39.56_{\pm2.85}$ | $93.28_{\pm0.42}$ | $59.74_{\pm3.22}$ | $47.26_{\pm5.19}$ | $84.26_{\pm1.80}$ |
> | HDS$^{ode}$ | $64.73_{\pm2.27}$ | $78.89_{\pm2.95}$ | $75.42_{\pm2.05}$ | $42.19_{\pm1.90}$ | $41.10_{\pm2.88}$ | $93.41_{\pm0.85}$ | $61.62_{\pm1.83}$ | $49.37_{\pm3.38}$ | $86.64_{\pm1.28}$ |

---

> ### Author Response · Authors · 2023-11-22
> **Response to Reviewer HBiR**
>
> Results on 1 training vertex each class.
>
> |    Model    |           Cora-CA | DBLP-CA            | News20             | IMDB4k-CA         | IMDB4k-CD         | DBLP4k-CC         | DBLP4k-CP         | Cooking            | NTU               |
> | :---------: | ----------------: | ------------------ | ------------------ | ----------------- | ----------------- | ----------------- | ----------------- | ------------------ | ----------------- |
> |     GCN     | $25.14_{\pm7.72}$ | $46.24_{\pm6.78}$  | $40.85_{\pm11.39}$ | $36.65_{\pm2.11}$ | $36.36_{\pm1.73}$ | $83.46_{\pm7.62}$ | $36.65_{\pm9.91}$ | $25.90_{\pm6.46}$  | $61.48_{\pm4.07}$ |
> |  GraphSAGE  | $25.21_{\pm3.91}$ | $46.92_{\pm10.31}$ | $42.44_{\pm12.20}$ | $36.23_{\pm0.83}$ | $36.10_{\pm1.17}$ | $77.55_{\pm8.27}$ | $37.13_{\pm8.01}$ | $26.68_{\pm5.91}$  | $60.83_{\pm6.53}$ |
> |    HGNN     | $27.35_{\pm2.53}$ | $47.88_{\pm6.49}$  | $46.63_{\pm10.01}$ | $37.28_{\pm1.97}$ | $37.12_{\pm1.73}$ | $93.32_{\pm0.85}$ | $42.15_{\pm9.86}$ | $28.83_{\pm7.05 }$ | $67.38_{\pm2.07}$ |
> |  HGNN$^+$   | $34.98_{\pm3.93}$ | $52.73_{\pm3.96}$  | $49.50_{\pm11.17}$ | $34.70_{\pm1.84}$ | $37.31_{\pm2.33}$ | $93.37_{\pm0.35}$ | $41.92_{\pm8.02}$ | $32.94_{\pm5.37}$  | $67.34_{\pm2.52}$ |
> |   UniGCN    | $27.94_{\pm1.31}$ | $50.91_{\pm6.86}$  | $47.08_{\pm9.65}$  | $37.02_{\pm1.70}$ | $36.83_{\pm1.47}$ | $92.73_{\pm1.15}$ | $39.83_{\pm8.70}$ | $28.25_{\pm7.16}$  | $66.60_{\pm3.37}$ |
> |   UniSAGE   | $38.80_{\pm2.04}$ | $53.33_{\pm3.14}$  | $49.23_{\pm9.24}$  | $37.05_{\pm1.68}$ | $36.93_{\pm1.53}$ | $92.64_{\pm2.54}$ | $42.39_{\pm7.13}$ | $35.26_{\pm5.00}$  | $67.96_{\pm2.96}$ |
> | HDS$^{ode}$ | $39.58_{\pm3.59}$ | $60.22_{\pm2.17}$  | $52.70_{\pm9.58}$  | $37.12_{\pm2.20}$ | $37.38_{\pm1.29}$ | $93.43_{\pm1.56}$ | $43.46_{\pm7.51}$ | $36.15_{\pm4.18}$  | $71.36_{\pm1.36}$ |
>
>
>
> **Cooking and NTU Dataset Introduction**. We briefly introduce the two datasets. The Cooking dataset consists of vertices representing dishes, with hyperedges indicating dishes that use the same ingredients. Each dish is also associated with categorical information indicating its cuisine type， such as French, Japanese. This dataset poses a unique challenge as it lacks initial vertex features, testing our model's ability to infer relationships solely based on hypergraph structure. The NTU dataset includes 3D shapes categorized into various classes like chairs, doors, etc. The vertex features are extracted using Multi-View Convolutional Neural Networks (MVCNN) [2] and Group-View Convolutional Neural Networks (GVCNN) [3] for 3D shapes. Since the NTU dataset does not come with an initial hypergraph structure, we constructed it by treating each 3D shape as a vertex and using a k-nearest neighbors method to build hyperedges based on MVCNN features and GVCNN features, respectively, thereby establishing the hypergraph structure. The following table shows the statistics of the Cooking dataset and NTU dataset. Since the Cooking dataset has no initial features, we initialize a unique feature for each vertex so that its degree is exactly the number of vertices.
>
>
>
> | Dataset | #Vertices | #Hyperedges | #Feature |
> | ------- | --------- | ----------- | -------- |
> | Cooking | 7,403     | 2,755       | 7,403    |
> | NTU     | 2,012     | 3,365       | 6,144    |

---

> ### Author Response · Authors · 2023-11-22
> **Response to Reviewer HBiR**
>
> **Improvements in Cooking and NTU**. In our experimental evaluation, we observe significant performance improvements in both the Cooking and NTU datasets when using our model. Specifically, when training with only 5 samples per class, our model achieves an enhancement of 3.13% on the Cooking dataset and 1.51% on the NTU dataset. With only 1 sample per class, the performance gains are even more pronounced, 2.52% on the Cooking dataset and a 5.00% on the NTU dataset. This indicates a notable enhancement in our model’s ability to capture and propagate limited supervisory information effectively across the hypergraph structure, even in scenarios with minimal training data. Moreover, in a more challenging few-shot scenario, these results highlight our model's proficiency in leveraging the overall hypergraph structure to extract and utilize latent relational information, especially when dealing with limited known vertex labels.
>
>
>
> In the case of the NTU dataset, where we construct the hypergraph structure from 3D shape features using MVCNN and GVCNN, the performance improvement emphasizes our model's capacity to exploit complex relational patterns from visually extracted features. Overall, these results not only validate the effectiveness of our dynamical system-based hypergraph neural network in diverse settings but also illustrate its potential to address challenges in hypergraph learning tasks where general hypergraph models may be limited.
>
>
>
> We hope this additional empirical evidence addresses your concerns and demonstrates the robustness and performance capabilities of our model in a variety of conditions.
>
>
>
> [1] Chen, Ding‐Yun, et al. "On visual similarity based 3D model retrieval." *Computer graphics forum*, 2003.
>
> [2] Su, Hang, et al. "Multi-view convolutional neural networks for 3d shape recognition." *Proceedings of the IEEE international conference on computer vision*, 2015.
>
> [3] Feng, Yifan, et al. "Gvcnn: Group-view convolutional neural networks for 3d shape recognition." *Proceedings of the IEEE conference on computer vision and pattern recognition*, 2018.

---

> ### Author Response · Authors · 2023-11-23
> **Inquiry and Gratitude Before Discussion Phase Conclusion**
>
> Dear Reviewer HBiR,
>
>
> We would like to express our gratitude for the time and effort you have dedicated to reviewing our paper. As the discussion phase is nearing its conclusion, we are eager to know if our responses have adequately addressed your concerns. If so, whether you could consider changing your rating.
>
>
> Please rest assured that we are completely dedicated to addressing any further questions or clarifications you may have before the end of the discussion. Your insights are invaluable to us, and we aim to incorporate them to enhance the quality of our work.
>
>
> Thank you again for your valuable contribution to the review process. We are looking forward to hearing from you soon.
>
>
> Best,
>
>
> The Authors

---

> > ### Comment · Reviewer_HBiR · 2023-11-23
> >
> > I want to thank the authors for addressing my concerns and conducting extra experiments. Nonetheless, the concept transforming graph/hypergraph's propagation process into continuous form is not a novel idea in this field. And I maintain my initial concern that the dynamical characteristics of the model could be validated only with dynamic, rather than statistic datasets.
> > Despite this concern, I appreciate the authors' efforts in providing a more comprehensive stability analysis, as well as the experimental results. Therefore I will update my score to 5.

---

> > > ### Author Response · Authors · 2023-11-23
> > > **Response to Reviewer HBiR**
> > >
> > > We would like to express our sincere gratitude for your reconsideration and are heartened to hear that our responses have addressed your concerns. Your acknowledgment of our efforts is greatly appreciated.
> > >
> > >
> > >
> > > Regarding the novelty of our approach, the concept of continuous neural networks is not entirely new. We believe that our work is a meaningful attempt to combine the concept of dynamical systems with hypergraph neural networks. While graphs are powerful tools, hypergraphs offer a more general and flexible framework, capable of transcending pairwise modeling with hyperedges that can connect more than two vertices. This capability allows hypergraphs to naturally model multi-modal correlations through their associative matrices, offering a more versatile approach than graphs. Hence, exploring the concept of continuity in hypergraph neural networks is essential to leverage the full potential of correlation structure.
> > >
> > >
> > >
> > > We acknowledge your concern regarding the validation of the model's dynamical characteristics, primarily using static rather than dynamic datasets. This indeed represents a limitation of most ODE-based methods, including graph ODEs. The availability of dynamic hypergraph datasets and the complexity involved in their study exceed the scope of our current work. However, your feedback provides a valuable direction for our future research. We are encouraged to explore dynamic datasets in future studies to further validate and enhance the dynamical aspects of our model.
> > >
> > >
> > >
> > > We would like to extend our sincere thanks for the upgrade in your evaluation score. And we thank you for your constructive feedback and for pointing us toward an exciting future research direction. We are committed to continuously improving our work and exploring new frontiers in the field.

---

### Official Review · Reviewer_KkAS · 2023-10-28

**Soundness:** 3 good
**Presentation:** 4 excellent
**Contribution:** 3 good
**Rating:** 5
**Confidence:** 3

**Summary:**

This paper theoretically introduces hypergraph dynamical systems based on a control-diffusion
ODE, which bridge hypergraphs and dynamical systems. It then proposes a neural implementation $HDS^{ode}$ and presents stability analysis of it and the connection to hypergraph neural networks. Finally, the paper empirically evaluates $HDS^{ode}$ using benchmark datasets and show its effectiveness to some extent. Some ablation studies are also included for more thorough investigation of the proposed method.

**Strengths:**

1.	Given the graph counterpart, it is a natural and interesting idea to develop a hypergraph neural ODE to improve the controllability and stabilization of information diffusion on hypergraphs. On the other hand, given that graphs and hypergraphs are different in nature, it is also a challenging problem how the system should be designed.
2.	The paper is well-written and easy to follow.
3.	The paper is very complete, with clear presentation of the method, some theoretical analysis and thorough empirical evaluation.

**Weaknesses:**

The main weakness of the paper is the weak empirical results supporting the effectiveness of the proposed method. It is unconvincing why small diffusion steps themselves constitute a real problem for hyper graph neural networks. Although $HDS^{ode}$’s performance does not suffer from more layers, $HDS^{ode}$ has very marginal improvement in terms of optimal performance. This is evident from all the experimental results (Figure 1, Table 1 and Table 2), where the improvement over baseline methods is barely noticeable and almost never statistically significant.

**Questions:**

"Dynamic systems" reads weird, and it is used kind of interchangeably with "dynamical systems" in the paper. Is there any specific reason to sometimes use "dynamic systems" instead of "dynamical systems" in the paper? If not, sticking with "dynamical systems" and "hypergraph dynamical systems" might be more appropriate. See discussion on Mathoverflow https://mathoverflow.net/questions/366856/why-is-a-dynamical-system-not-a-dynamic-system" about why the right mathematical jargon is "dynamical systems".

---

> ### Author Response · Authors · 2023-11-17
> **Response to Reviewer KkAS**
>
> We thank the reviewer for the insightful comments and feedback. We hope that our response can address your concerns.
>
> $\newline$
>
> **Response to Weakness**
>
> Thank you for your critical assessment of our empirical results. We acknowledge your concerns regarding the performance improvement demonstrated in our results. However, it is important to emphasize that the **primary focus** of our work is addressing a common and significant challenge in (hyper)graph neural networks (that is, the notable decrease in performance with an increasing number of layers) [1,2].
>
>
>
> **We wish to highlight the dual aspects of performance** in our work: not only the performance metrics but also the stability of performance. In practical applications of hypergraph neural networks, the specific dependencies required for optimal performance of many existing methods may be challenging to meet, since they depend on specific conditions. Therefore, our emphasis on **robustness** and reduced dependence ensures reliable and consistent performance across various scenarios and layer depths. This is why the robustness becomes particularly valuable. Our model's ability to maintain stability and consistency at increased layer depths addresses a fundamental limitation. Our model not only tackles the stability issues with increased layers but also provides performance enhancements.
>
>
>
> The performance improvement stems from three main factors.
>
> - **Continuous Representations**. The adoption of an ODE-based model in HDS$^{ode}$ allows for smoother dynamics in the hidden layers [3], promoting more effective learning and representation.
> - **Extended Interaction Range**. General hypergraph neural networks are often limited to 2 layers, restricting vertices to interact only with 1-hop and 2-hop neighbors. HDS$^{ode}$'s diffusion mechanism effectively extends this range, enabling interactions across the entire hypergraph [4]. This broader scope of interaction is instrumental in capturing more complex and global relationships within the data, thereby improving performance.
> - **Fine-Tuning Through Control Term**. A distinctive part of our model is the inclusion of a control term, which allows for the fine-tuning of representations during diffusion. This aspect is not commonly found in other hypergraph neural network models. The control term acts as an auxiliary function, fine-tuning the diffusion process to better align with the specific dataset. This capability adds an additional layer of adaptability to our model, further enhancing its performance.
>
> $\newline$
>
> **Response to Question**
>
> We appreciate your insightful comment regarding the terminology of "dynamic system" versus "dynamical system". After reflecting on your feedback, we recognize the importance of consistent and precise terminology in our field. Given that our model represents a new form of hypergraph neural network with a focus on the study of system dynamics, we agree that "dynamical system" is the more appropriate and established term in mathematical terminology. We are in the process of revising our manuscript to consistently use "dynamical system" throughout, ensuring alignment with standard terminology and enhancing the clarity of our work.
>
> $\newline$
>
> Thank you once again for your valuable time and consideration. We hope our response addresses your concerns.
>
> $\newline$
>
> [1] Qimai Li, Zhichao Han, Xiao-Ming Wu. "Deeper insights into graph convolutional networks for semi-supervised learning." *Proceedings of the AAAI conference on artificial intelligence*, 2018.
>
> [2] Deli Chen, Yankai Lin, Wei Li, Peng Li, Jie Zhou, Xu Sun. "Measuring and relieving the over-smoothing problem for graph neural networks from the topological view." *Proceedings of the AAAI conference on artificial intelligence*, 2020.
>
> [3] Xhonneux, Louis-Pascal, Meng Qu, and Jian Tang. "Continuous graph neural networks." *International Conference on Machine Learning*, 2020.
>
> [4] Johannes Gasteiger, Stefan Weißenberger, and Stephan Günnemann. "Diffusion improves graph learning." *Advances in neural information processing systems*, 2019.

---

> ### Comment · Reviewer_KkAS · 2023-11-21
>
> Thank the authors for the response. However, I am not convinced by the argument that since robustness is the main focus, the weak improvement can be well justified. In particular, to justify the claim such as "this broader scope of interaction is instrumental in capturing more complex and global relationships within the data, thereby improving performance", one would expect that the proposed method is not only robust, but also improves the performance significantly, because of the capability to capture additional information.
>
> The current empirical results just imply that more information beyond 1-2 hops has very limited value, at least in the datasets and tasks demonstrated in this paper. If that is the case, previous methods, despite being shallow, suffice.

---

> > ### Author Response · Authors · 2023-11-22
> > **Response to Reviewer KkAS (Part 1/3)**
> >
> > We thank for your feedback on our work. We understand your concerns regarding the performance improvement in our experiments, and we add an additional ablation study that we have conducted to further validate our model's performance. In difficult scenarios, the hypergraph structure better mine the complex relationships behind the data and better solve problems.
> >
> >
> >
> > Under conditions with limited effective supervisory information, such information cannot be sufficiently propagated through the network, resulting in limitations of general hypergraph neural networks. However, our ODE-based model, through the use of more layers, allows for the more extensive propagation of this limited supervisory information. This more comprehensive message passing enables our model to overcome the performance bottlenecks with the propagation capabilities of general hypergraph neural networks (with average 0.95 and 1.90 accuracy enhancement in 5 and 1 training vertex each class, respectively). In our experiments, we have explored scenarios with significantly fewer training samples each class, specifically, datasets with only 5 and even 1 training vertex each class. These few-shot conditions present a more challenging environment for learning accurate vertex representations, as the available information is considerably limited. The results of these experiments are presented in the following table.
> >
> >
> >
> > Our model shows a more obvious advantage over general hypergraph neural network methods. This improvement becomes clearer in these few-shot scenarios. The reason for this can be attributed to the fact that with fewer training vertices, the need to capture information beyond 1-2 hops becomes more critical since fewer labels within the neighbor range with low distance. Our model's ability to leverage a broader scope of interaction allows it to access and utilize more comprehensive information, which is especially beneficial in situations where known vertex information is scarce.
> >
> >
> >
> > Our model shows greater potential for hypergraph inference tasks with limited known vertex information, indicating its capacity to achieve superior performance in more challenging and realistic scenarios. Our model's broader interaction scope is instrumental in capturing more complex and global relationships within the data, thereby enhancing performance.
> >
> >
> >
> > In addition to the previously mentioned experiments, we show the performance of our model in diverse scenarios. Specifically, we have included tests on datasets with unique characteristics, without initial vertex features (Cooking Dataset), and without an initial hypergraph structure (NTU Dataset [1]). The Cooking dataset consists of vertices representing dishes, with hyperedges indicating dishes that use the same ingredients. Each dish is also associated with categorical information indicating its cuisine type. This dataset poses a unique challenge as it lacks initial vertex features, testing our model's ability to infer relationships solely based on hypergraph structure. The NTU dataset includes 3D shapes categorized into various classes. The vertex features are extracted using Multi-View Convolutional Neural Networks (MVCNN) [2] and Group-View Convolutional Neural Networks (GVCNN) [3] for 3D shapes. Since the NTU dataset does not come with an initial hypergraph structure, we constructed it by treating each 3D shape as a vertex and using a k-nearest neighbors method to build hyperedges based on features, thereby establishing the hypergraph structure. Our experiments on these datasets demonstrate that our model outperforms standard hypergraph neural network approaches even in scenarios lacking initial vertex features or hypergraph structures. These results further validate the capability of our model to adapt in various complex scenarios.
> >
> >
> >
> > We hope this additional empirical evidence addresses your concerns and demonstrates the robustness and performance capabilities of our model in a variety of conditions.

---

> > ### Author Response · Authors · 2023-11-22
> > **Response to Reviewer KkAS (Part 2/3)**
> >
> > Results on 10 training vertex each class (reported in our paper).
> >
> > |    Model    |       Cora-CA       |       DBLP-CA       |       News20        |      IMDB4k-CA      |      IMDB4k-CD      |      DBLP4k-CC      |      DBLP4k-CP      |
> > | :---------: | :-----------------: | :-----------------: | :-----------------: | :-----------------: | :-----------------: | :-----------------: | :-----------------: |
> > |     GCN     |  $65.99_{\pm3.69}$  |  $82.22_{\pm1.05}$  |  $67.57_{\pm0.70}$  |  $43.47_{\pm2.39}$  |  $41.02_{\pm2.22}$  |  $90.18_{\pm1.22}$  |  $64.47_{\pm0.90}$  |
> > |  GraphSAGE  |  $66.44_{\pm2.82}$  |  $81.07_{\pm1.50}$  |  $69.59_{\pm0.89}$  |  $42.05_{\pm1.95}$  |  $41.07_{\pm2.11}$  |  $92.18_{\pm0.38}$  |  $64.34_{\pm1.58}$  |
> > |    HGNN     |  $67.58_{\pm1.83}$  |  $82.83_{\pm1.09}$  |  $76.58_{\pm0.94}$  |  $43.21_{\pm2.39}$  |  $41.08_{\pm2.43}$  |  $93.46_{\pm0.77}$  |  $67.99_{\pm2.12}$  |
> > |  HGNN$^+$   |  $66.85_{\pm2.24}$  |  $82.40_{\pm1.27}$  |  $76.49_{\pm1.30}$  |  $43.74_{\pm1.42}$  |  $41.49_{\pm2.54}$  |  $93.46_{\pm1.09}$  |  $68.76_{\pm2.73}$  |
> > |   UniGCN    |  $66.47_{\pm2.04}$  |  $82.36_{\pm1.09}$  |  $76.56_{\pm1.21}$  |  $43.34_{\pm3.26}$  |  $41.33_{\pm2.50}$  |  $93.28_{\pm0.87}$  |  $67.68_{\pm1.90}$  |
> > |   UniSAGE   |  $68.59_{\pm1.61}$  |  $82.16_{\pm1.25}$  |  $75.52_{\pm1.22}$  |  $42.82_{\pm2.66}$  |  $41.62_{\pm3.05}$  |  $93.64_{\pm0.58}$  |  $67.81_{\pm2.12}$  |
> > | HDS$^{ode}$ | ${68.92_{\pm1.28}}$ | ${83.05_{\pm0.53}}$ | ${76.75_{\pm1.07}}$ | ${44.26_{\pm2.11}}$ | ${42.30_{\pm2.92}}$ | ${93.85_{\pm0.50}}$ | ${69.52_{\pm1.19}}$ |
> >
> > $\newline$
> >
> > Results on 5 training vertex each class.
> >
> > |    Model    |           Cora-CA |           DBLP-CA |            News20 | IMDB4k-CA         | IMDB4k-CD         | DBLP4k-CC         | DBLP4k-CP         | Cooking           | NTU               |
> > | :---------: | ----------------: | ----------------: | ----------------: | ----------------- | ----------------- | ----------------- | ----------------- | ----------------- | ----------------- |
> > |     GCN     | $54.61_{\pm4.64}$ | $73.65_{\pm6.84}$ | $62.81_{\pm2.65}$ | $39.14_{\pm2.97}$ | $38.54_{\pm2.44}$ | $89.46_{\pm1.17}$ | $56.96_{\pm5.85}$ | $32.87_{\pm3.99}$ | $79.64_{\pm1.85}$ |
> > |  GraphSAGE  | $53.11_{\pm3.74}$ | $73.58_{\pm5.40}$ | $62.66_{\pm3.73}$ | $39.72_{\pm1.16}$ | $38.51_{\pm1.34}$ | $91.08_{\pm1.03}$ | $55.95_{\pm5.33}$ | $32.73_{\pm5.11}$ | $76.78_{\pm1.13}$ |
> > |    HGNN     | $60.52_{\pm4.12}$ | $75.55_{\pm4.27}$ | $75.05_{\pm1.57}$ | $40.63_{\pm1.76}$ | $39.42_{\pm2.58}$ | $93.35_{\pm0.77}$ | $58.07_{\pm6.02}$ | $45.35_{\pm6.71}$ | $85.35_{\pm1.62}$ |
> > |  HGNN$^+$   | $58.17_{\pm5.10}$ | $77.34_{\pm2.49}$ | $75.43_{\pm1.81}$ | $40.92_{\pm2.92}$ | $39.41_{\pm2.58}$ | $93.03_{\pm1.46}$ | $60.83_{\pm4.70}$ | $47.87_{\pm4.21}$ | $85.27_{\pm0.92}$ |
> > |   UniGCN    | $58.36_{\pm5.11}$ | $74.54_{\pm4.88}$ | $74.98_{\pm1.70}$ | $41.51_{\pm2.56}$ | $39.94_{\pm2.31}$ | $93.04_{\pm1.28}$ | $59.91_{\pm4.92}$ | $45.98_{\pm7.12}$ | $84.10_{\pm0.86}$ |
> > |   UniSAGE   | $62.56_{\pm3.27}$ | $77.94_{\pm2.39}$ | $73.90_{\pm2.85}$ | $41.67_{\pm2.50}$ | $39.56_{\pm2.85}$ | $93.28_{\pm0.42}$ | $59.74_{\pm3.22}$ | $47.26_{\pm5.19}$ | $84.26_{\pm1.80}$ |
> > | HDS$^{ode}$ | $64.73_{\pm2.27}$ | $78.89_{\pm2.95}$ | $75.42_{\pm2.05}$ | $42.19_{\pm1.90}$ | $41.10_{\pm2.88}$ | $93.41_{\pm0.85}$ | $61.62_{\pm1.83}$ | $49.37_{\pm3.38}$ | $86.64_{\pm1.28}$ |

---

> > ### Author Response · Authors · 2023-11-22
> > **Response to Reviewer KkAS (Part 3/3)**
> >
> > Results on 1 training vertex each class.
> >
> > |    Model    |           Cora-CA | DBLP-CA            | News20             | IMDB4k-CA         | IMDB4k-CD         | DBLP4k-CC         | DBLP4k-CP         | Cooking            | NTU               |
> > | :---------: | ----------------: | ------------------ | ------------------ | ----------------- | ----------------- | ----------------- | ----------------- | ------------------ | ----------------- |
> > |     GCN     | $25.14_{\pm7.72}$ | $46.24_{\pm6.78}$  | $40.85_{\pm11.39}$ | $36.65_{\pm2.11}$ | $36.36_{\pm1.73}$ | $83.46_{\pm7.62}$ | $36.65_{\pm9.91}$ | $25.90_{\pm6.46}$  | $61.48_{\pm4.07}$ |
> > |  GraphSAGE  | $25.21_{\pm3.91}$ | $46.92_{\pm10.31}$ | $42.44_{\pm12.20}$ | $36.23_{\pm0.83}$ | $36.10_{\pm1.17}$ | $77.55_{\pm8.27}$ | $37.13_{\pm8.01}$ | $26.68_{\pm5.91}$  | $60.83_{\pm6.53}$ |
> > |    HGNN     | $27.35_{\pm2.53}$ | $47.88_{\pm6.49}$  | $46.63_{\pm10.01}$ | $37.28_{\pm1.97}$ | $37.12_{\pm1.73}$ | $93.32_{\pm0.85}$ | $42.15_{\pm9.86}$ | $28.83_{\pm7.05 }$ | $67.38_{\pm2.07}$ |
> > |  HGNN$^+$   | $34.98_{\pm3.93}$ | $52.73_{\pm3.96}$  | $49.50_{\pm11.17}$ | $34.70_{\pm1.84}$ | $37.31_{\pm2.33}$ | $93.37_{\pm0.35}$ | $41.92_{\pm8.02}$ | $32.94_{\pm5.37}$  | $67.34_{\pm2.52}$ |
> > |   UniGCN    | $27.94_{\pm1.31}$ | $50.91_{\pm6.86}$  | $47.08_{\pm9.65}$  | $37.02_{\pm1.70}$ | $36.83_{\pm1.47}$ | $92.73_{\pm1.15}$ | $39.83_{\pm8.70}$ | $28.25_{\pm7.16}$  | $66.60_{\pm3.37}$ |
> > |   UniSAGE   | $38.80_{\pm2.04}$ | $53.33_{\pm3.14}$  | $49.23_{\pm9.24}$  | $37.05_{\pm1.68}$ | $36.93_{\pm1.53}$ | $92.64_{\pm2.54}$ | $42.39_{\pm7.13}$ | $35.26_{\pm5.00}$  | $67.96_{\pm2.96}$ |
> > | HDS$^{ode}$ | $39.58_{\pm3.59}$ | $60.22_{\pm2.17}$  | $52.70_{\pm9.58}$  | $37.12_{\pm2.20}$ | $37.38_{\pm1.29}$ | $93.43_{\pm1.56}$ | $43.46_{\pm7.51}$ | $36.15_{\pm4.18}$  | $71.36_{\pm1.36}$ |
> >
> >
> >
> >
> >
> > [1] Chen, Ding‐Yun, et al. "On visual similarity based 3D model retrieval." *Computer graphics forum*, 2003.
> >
> > [2] Su, Hang, et al. "Multi-view convolutional neural networks for 3d shape recognition." *Proceedings of the IEEE international conference on computer vision*, 2015.
> >
> > [3] Feng, Yifan, et al. "Gvcnn: Group-view convolutional neural networks for 3d shape recognition." *Proceedings of the IEEE conference on computer vision and pattern recognition*, 2018.

---

> > ### Author Response · Authors · 2023-11-22
> > **Addtional Interpretation of Datasets and Results**
> >
> > The following table shows the statistics of the Cooking dataset and NTU dataset. Since the Cooking dataset has no initial features, we initialize a unique feature for each vertex so that its degree is exactly the number of vertices.
> >
> >
> >
> > | Dataset | #Vertices | #Hyperedges | #Feature |
> > | ------- | --------- | ----------- | -------- |
> > | Cooking | 7,403     | 2,755       | 7,403    |
> > | NTU     | 2,012     | 3,365       | 6,144    |
> >
> >
> >
> > In our experimental evaluation, we observe significant performance improvements in both the Cooking and NTU datasets when using our model. Specifically, when training with only 5 samples per class, our model achieves an enhancement of 3.13% on the Cooking dataset and 1.51% on the NTU dataset. With only 1 sample per class, the performance gains are even more pronounced, 2.52% on the Cooking dataset and a 5.00% on the NTU dataset. This indicates a notable enhancement in our model’s ability to capture and propagate limited supervisory information effectively across the hypergraph structure, even in scenarios with minimal training data. Moreover, in a more challenging few-shot scenario, these results highlight our model's proficiency in leveraging the overall hypergraph structure to extract and utilize latent relational information, especially when dealing with limited known vertex labels.
> >
> >
> >
> > In the case of the NTU dataset, where we construct the hypergraph structure from 3D shape features using MVCNN and GVCNN, the performance improvement emphasizes our model's capacity to exploit complex relational patterns from visually extracted features. Overall, these results not only validate the effectiveness of our dynamical system-based hypergraph neural network in diverse settings but also illustrate its potential to address challenges in hypergraph learning tasks where general hypergraph models may be limited.

---

> ### Author Response · Authors · 2023-11-23
> **Inquiry and Gratitude Before Discussion Phase Conclusion**
>
> Dear Reviewer KkAS,
>
>
> We would like to express our gratitude for the time and effort you have dedicated to reviewing our paper. As the discussion phase is nearing its conclusion, we are eager to know if our responses have adequately addressed your concerns. If so, whether you could consider changing your rating.
>
>
> Please rest assured that we are completely dedicated to addressing any further questions or clarifications you may have before the end of the discussion. Your insights are invaluable to us, and we aim to incorporate them to enhance the quality of our work.
>
>
> Thank you again for your valuable contribution to the review process. We are looking forward to hearing from you soon.
>
>
> Best,
>
>
> The Authors

---

### Official Review · Reviewer_4j9N · 2023-10-30

**Soundness:** 4 excellent
**Presentation:** 4 excellent
**Contribution:** 3 good
**Rating:** 8
**Confidence:** 5

**Summary:**

In this paper, the authors target on the task of representation learning with high-order correlations on hypergraph, in which the challenge is the sub-optimal problem during the process the neural network. Existing hypergraph neural networks cannot be deeper than 2 layers and is unstable. This problem is a common but challenged issue in this field. The authors introduce the framework of hypergraph dynamic systems, which connects hypergraph learning and dynamic systems to achieve continuous dynamics of representation using high-order correlations. The authors further propose an implementation of hypergraph dynamic systems based on ordinary differential equation and experiments have shown stable and satisfied performance. This control-diffusion process introduced in this paper have been demonstrated effectiveness through the results and theoretical discussions.

**Strengths:**

This paper targets on an important but challenged task in representation learning, i.e., how to achieve stable representation learning in hypergraph neural network, which is also a common issue in the general graph neural networks. Usually, HGNNs cannot be more than 2 layers, which leads to performance degradation significantly. The introduced hypergraph dynamic systems framework in this paper bridges hypergraph learning and dynamic systems, which can take the advantages of hypergraph on high-order correlation modeling and dynamic systems on controllable diffusion process. The idea of hypergraph dynamic systems is novel. It is a good attempt towards better representation learning and could be helpful to a broad field.

The authors also propose an implementation of HDS using ODE, and a multi-layer HDS-ode is given. The stability analysis has also been detailed analyzed. The difference between HDS-ode and traditional HGNNs has discussed.

Experiments are sufficient. Experiments on semi-supervised vertex classification with two different settings have been conducted on 7 datasets. Experimental results have clearly shown the superior performance of HDS-ode compared with recent state-of-the-art GNN/HGNN methods. From the results, we can observe the control-diffusion process of HDS-ode is stable, which solves the limitations of existing HGNNs, i.e. only 1 or 2 layers can be used.

In general, this paper is well organized and writing. The related works are sufficient and the motivation is clear. The method has been detailed introduced. This paper brings in a new aspect of representation learning of taking both high-order correlation modeling and dynamic systems into consideration simultaneously, which has the potential to have broad impact.

**Weaknesses:**

For the framework figure (Fig. 2), a more detailed and clearer introduction should be helpful.

There are a few typos. Please find and correct them.

**Questions:**

1. For the framework figure (Fig. 2), a more detailed and clearer introduction should be helpful.
2. As shown in Fig. 1, the performance of HDS-ode increases during the first 8 layers and becomes stable then. Can the authors further explain how to control the diffusion speed during this procedure?
3. There are a few typos. Please find and correct them.

**Details Of Ethics Concerns:**

No ethics concerns.

---

> ### Author Response · Authors · 2023-11-17
> **Response to Reviewer 4j9N**
>
> We thank the reviewer for your valuable comments and suggestions. We are currently revising the manuscript to provide a more detailed and comprehensible description of the framework. Additionally, we are reviewing the paper to identify and correct all typos to enhance the overall clarity and readability of our work.
>
> $\newline$
>
> **Response to Question 2**
>
> Thank you for your question regarding the control of diffusion speed. Each layer in HDS$^{ode}$ includes a diffusion term and a control term. The hyperparameters $\alpha_v,\alpha_e$ in the diffusion matrix $\begin{bmatrix}
>             -\alpha_v I& \alpha_v D_v^{-1}H \newline
>             \alpha_e D_e^{-1}H^\top &-\alpha_e I
>         \end{bmatrix}$ can be considered as the speed in the diffusion process. Taking vertex representations for explanation, $\alpha_v$ represents the proportion of vertex representations in each layer obtained from hyperedges in the diffusion process.
>
>
>
> Expanding the vertex representation of Eq. (7), we get $X_v(t+1)=(1-\alpha_v)X_v(t+\frac{1}{2})+\alpha_v D_v^{-1}HX_e$. Specifically,  $\alpha_v$ dictates the proportion of vertex representations derived from hyperedges during diffusion. For instance, with a lower $\alpha_v$ value, close to 0, the diffusion speed is minimized, and the vertex representation $X_v(t+1)$ remains closer to its previous state $X_v(t+\frac{1}{2})$. Conversely, as $\alpha_v$ approaches 1, $X_v(t+1)$ primarily consists of hyperedge representations $D_v^{-1}HX_e$.
>
>
>
> In our experimental setup, $\alpha_v,\alpha_e$  are kept constant, implying a consistent diffusion speed across layers. When the number of layers increases (for example, reaching 8 layers), the representation of vertices and hyperedges is stable after diffusion, and the performance no longer increases significantly.
>
> $\newline$
>
> Thanks again for your time and consideration. We hope our response addresses your concerns.

---

### Official Review · Reviewer_3YqS · 2023-11-05

**Soundness:** 3 good
**Presentation:** 3 good
**Contribution:** 2 fair
**Rating:** 6
**Confidence:** 4

**Summary:**

This paper introduces hypergraph dynamic systems (HDS) to characerize the continuous dynamics of representations, and then proposes a control-diffusion HDS by an ODE. A multi-layer HDS-ODE is designed as a neural implementation, having the properties of controllability and stabilization, and   can capture long-range correlations among vertices. The paper performs evaluation experiments on 7 datsets to show its dominant performance.

**Strengths:**

1. The pape present the implementation of HDS-ODE framework by posing the neural implementation of the control step and diffusion step in sequence  via Lie-Trotter splitting method.
2. The time complexity and relation to HGNN+ are analyzed. And the experiments show its very good performance.
3. The properties of HDS-ODE are discussed and we appreciate such an effort on analysis in theory, although we do have concerns on these contents (please see comments below).

**Weaknesses:**

1. To the terminology (and the preliminary math tools) of dynamical systems used in this paper, it seems that the authors does not soundly cook its article based on the strict math that has been widely accepted in math and engineering. See Queation 1, Weakness point 2 and 3.

2. The discussion of statibility seems wrong. That's why we say the authors may not well pick up knowledge of (linear/nonlinear) dynamical systems. Sec. 5.1 discussed the stability of HDS-ODE, where the statement of the first sentence in this section is basically wrong. The so-called "control" term also iterate over time. It cannot be simplified the stability analysis of HDS-ODE as simply a dissusion of linear system X_dot = A X. Let us use the eq.(3) to clarify the point simply. Supposing we accept the split of the general state-space equation X_dot = f(X, t) as eq.(3), it is obviously that the stability analysis is a general stability analysis of nonlinear system, besides AX(t) there also exists g(X(t))! If you said your discussion of HDS-ODE refer to the neural implementation, we can see that in eq. (6) the nonlinear term is still there, except it is in NN form, and then embeds into eq.(7) to complete the iteration t+1. Furthermore, these propositions on stability for your "simplified" linear systems are well-known in the field of electrical engineering.

3. Your abstract and contribution summary tell that yours studies the "controllability", which we could find anything related to it. And regarding the starting point of the most general eq.(2), there cannot be any controllablity-related problem can be formulated. Maybe you refer to something different? We strongly recommend the authors to learn essentials of dynamical systems, it helps to avoid conceptual misunderstandings and misuse for better communications.

   To help you with essential knowledge on dynamical systems, you may refer to the following classic textbooks (basics on linear, nonlinear systems):

   - Zhou, K., Doyle, J. C., & Glover, K. (1995). Robust and Optimal Control. Pearson.
   - H. K. Khalil, “Nonlinear Systems,” 3rd Edition, Prentice Hall, Upper Saddle River, 2002.

Your idea may be valuable and appreciated, considering your sound performance in experiments. However, you really have to first carefully deal with theory and fix any possible mistakes.

**Questions:**

1. Why do you call these two terms in eq.(3) as the "control" term and the "diffusion" term. We are not familiar with the terminology in the "small" field (that is consisted of these ~5 papers in introduction). However, in the mature fields of "dynamical systems" in math, "stochastic analysis" in probability or mathematical finance  and "control theory/engineering" or "cybernetics" in engineering, neither the name of "control" nor "diffusion" may be  properly defined. The control term usually refers to external signals or extrogeneous variables (in econometrics) that can be designed or modified. Indeed, if you assume, eq.(3) is not an autonomous system in nature (if purely looking at (3) itself, it is), but the feedback system by using the nonlinear state feedback law g(X) as the control "u" for the linear dynamical system X_dot = A X + u. It somehow legitimate the name "control". However, it is still not recommended in this way since it is confusing. Referring to the diffusion term, we cannot see why it can be call as this name, since the diffusion term is h(X)dW in the SDE dX(t) = f(X, t) dt + h(X,t) dW, where W is the Brownian motion. The AX(t) term in eq.(3) is actually the term of f(X,t), so the drift term.
2. See Weakness point 2 on Sec.5.1. Please explain, in particular, the first sentence of Sec.5.1, which seems not correct.

---

> ### Author Response · Authors · 2023-11-17
> **Response to Reviewer 3YqS (Part 1/2)**
>
> We thank the reviewer for the insightful comments and feedback.
>
> We apologize for the lack of clarity in our manuscript and appreciate the opportunity to elucidate our approach. Our method references some methods from dynamical systems to address a common and significant challenge of a decrease in performance with an increasing number of layers in hypergraph neural networks. **It's important to clarify that our focus is on enhancing hypergraph neural networks by dynamical systems concepts, rather than developing a dynamical system.** Considering your valuable feedback, we believe that renaming our model to "Hypergraph Dynamic Neural Network" would more accurately represent the essence of our work. Our proposed model is an enhancement of general hypergraph neural networks, designed to be a next-generation framework within this domain. Our model significantly enhances the stability of existing hypergraph neural networks. This advanced framework effectively combines the correlation hypergraph structure properties with the dynamic aspects of neural networks, offering a more comprehensive model than general hypergraph neural networks.
>
>
>
> **Response to Weakness 2, Question 2**
>
> Thank you for your feedback on the stability discussion in our paper.
>
>
>
> In our analysis, we primarily focus on the stability of the diffusion component, which is the main element of our model and a critical aspect of hypergraph neural networks in general. This is due to the diffusion component's significant role in determining the overall effectiveness of the hypergraph model. To clarify, the first sentence in Section 5.1 is intended to convey that diffusion in our model represents the teleportation of vertex and hyperedge representations over time. As this diffusion process is a main component of our ODE-based approach, its stability analysis is crucial.
>
>
>
> The control term in our model is designed to act as a secondary, fine-tuning term. Its primary function is to add minor perturbations to the main diffusion process, allowing the model to adapt more effectively to different data. The diffusion component dictates the primary dynamics of the network, the control term provides the minor necessary adjustments to enhance performance and adaptability without fundamentally altering the core stability characteristics of the model. That's the reason why we mainly analyze the stability of diffusion.
>
>
>
> If the parameters in the function $g$ are too large, they amplify $X$, thereby affecting the main diffusion component. By controlling the parameter size of $W$ and $b$, we ensure that the transformations they induce in the representation space are bounded. In our neural network implementation, we use L2 regularization to keep the values of $W$ and $b$ within a lower range. This helps control the size of these parameters, thereby preventing potential instability that large weights may cause. This approach is a standard practice in neural network training, suitable for our system architecture, and helps ensure the boundedness of the model's output, maintaining stability.
>
>
>
> **Response to Weakness 3**
>
> We apologize for the confusion caused by our use of the term "control" in our paper. We realize that this term may carry different connotations in various fields.  We are in the process of revising our manuscript to enhance the clarity of our work. In our research, "control" refers to the process of fine-tuning or adjusting the diffusion in our hypergraph neural network model. This usage of the word is distinct from the concept of "controllability" in control theory.
>
>
>
> In our model, "controllability" refers to the capability to fine-tune and adjust the diffusion process. Diffusion or vertex-(hyper)edge-vertex representation teleportation process is the central aspect of most graph and hypergraph neural networks. Traditionally, these networks primarily focus on representing diffusion by vertex-(hyper)edge-vertex teleportation process without additional fine-tuning. Our model includes a learnable function $g$, which acts as an auxiliary element. The function is specifically designed to fine-tune and control the primary diffusion component. By incorporating this learnable control function, we enhance the model's ability to adapt more precisely to the unique features of the hypergraph data. The "controllability" refers to our model's enhanced capability to refine the diffusion process through function $g$, which marks a significant enhancement over general (hyper)graph neural networks.

---

> ### Author Response · Authors · 2023-11-17
> **Response to Reviewer 3YqS (Part 2/2)**
>
> **Response to Question 1**
>
> Thank you for your insightful comments regarding the terminology used in our paper. In the field of graph and hypergraph, particularly when discussing (hyper)graph neural networks, "diffusion" is commonly used to describe the process by which features or representations teleport across the vertices of the graph or across the vertices and hyperedges of the hypergraph. This usage aligns with the intuitive notion of diffusion as a teleport process. In our model, the term "diffusion" refers to the $AX$ term in Eq. 3, representing the spread of representations across the hypergraph's vertices and hyperedges with a similar definition as existing methods [1-5].
>
>
>
> In our model, the term "control" specifically refers to a fine-tuning step that complements the primary diffusion process. It acts as an auxiliary function, adjusting and controlling the diffusion $AX$ to align more precisely with the downstream goals. This control step is internal, and designed to refine the diffusion of representations across the hypergraph's vertices and hyperedges, enhancing the performance and accuracy of the diffusion. By integrating this fine-tuning control function, our model achieves a balance between diffusion of vertex and hyperedge representations and task-aware adjustments, leading to more reliable and accurate performance.
>
>
>
> We hope our response addresses your concerns. We also welcome any new questions you may have.
>
>
>
> [1] Johannes Gasteiger, Stefan Weißenberger, and Stephan Günnemann. "Diffusion improves graph learning." *Advances in neural information processing systems*, 2019.
>
> [2] Xhonneux, Louis-Pascal, Meng Qu, and Jian Tang. "Continuous graph neural networks." *International Conference on Machine Learning*, 2020.
>
> [3] Yifei Wang, Yisen Wang, Jiansheng Yang, Zhouchen Lin. "Dissecting the diffusion process in linear graph convolutional networks." *Advances in Neural Information Processing Systems*, 2021.
>
> [4] Ben Chamberlain, James Rowbottom, Maria I Gorinova, Michael Bronstein, Stefan Webb, Emanuele Rossi. "Grand: Graph neural diffusion." *International Conference on Machine Learning*, 2021.
>
> [5] Peihao Wang, Shenghao Yang, Yunyu Liu, Zhangyang Wang, Pan Li. "Equivariant Hypergraph Diffusion Neural Operators." *International Conference on Learning Representations*, 2022.

---

> ### Comment · Reviewer_3YqS · 2023-11-21
>
> We thank the authors for their detailed explanations and responses.
>
> **What we have been convinced or we agree with**:
>
> We got your whole story of your topic and what you really mean by terms like "control term", "controllability", etc. And indeed your work is not developing a dynamical model on hypergraphs, but enhancing a better hypergraph neural networks. Actually, what you have responded, as quoted below, soundly explained why we call it "control" term.
> > The control term in our model is designed to act as a secondary, fine-tuning term.
>
> This follows the naming convention by considering the control term as a feedback control law on state variable $x$. However, you are suggested to provide more description on the "control" term (e.g., that can be fine-tuned, or explicitly, as $u(t)$ first then $u(t) = g(X(t))$, etc.) to clarify confusions. For the "controllability", actually what you described (as quote below) is exactly covered/described by this term.
> > "controllability" refers to the capability to fine-tune and adjust the diffusion process.
>
> However, it has a strict definition, which is somehow simple in LTI systems while fairly complicated in nonlinear systems. If you like to use this term, you have to follow its definition and reason/analyze it strictly in math, which may not be an easy task. Otherwise, you may consider using an alternative word.
> Although people in CS on diffusion NN may not be quite strict with terminology from math (dynamical systems) or control theory, the early papers on diffusion model (NN) comply with naming convention and can be strictly explained in math. We strongly suggest the authors to be carefully deal with them to avoid any possible confusions.
>
> **The main issue that concerns the reviewer**:
>
> For the stability, sorry, you still need to pick up a bit on stability on feedback systems (for nonlinear dynamical systems). This is a serious topic/question you have to address carefully. You cannot careless say the secondary control term is minor in value (which seems not according to your description) or fine-tuned later, then ignore this term and oversimplify the analysis of the stability of the whole nonlinear system as an LTI system.  For ODEs / dynamical systems, you cannot simply say that the term is or is kept small in fine-tuning and thus we don't need to consider it in stability analysis and the stability result is just the classic one for LTI systems. This key lesson we have learn for dynamical systems, in a comparison with regular algebraic-equation models. Well, if this control term is small in value, FYI referring to some results from optimal/robust control (the reviewer cannot name specific results, but we believe there exist several useful theorems/propositions), we are able to gain the stability of the whole system given certain conditions on the (uniform) bounds of this terms.  Well, your case seems a bit more general. No matter which case you are dealing with, you have to take serious the "stability" analysis for ODE  and address it rigorously. You may provide far stronger conditions that guarantee the stability than what you need in general in applying the model, which is common since theoretical analysis is hard and usually cannot be done at one-shot.
>
> Overall, we like to upgrade our score to 3 or 5 up to what you have explained. Again, if you like to improve your paper further to an acceptable level, in the reviewer's perspective, you need to **improve your stability analysis and address it rigorously in math**, as you also have said *its stability analysis is crucial*. What you have presented, **to our view, it is not a niche but an error** (which may not matter for some CSers, while leaving such an obvious misunderstanding in math may not be acceptable for ICLR). This is an interesting work, and thus please pick up stability analysis for ODEs and fix it.

---

> > ### Author Response · Authors · 2023-11-22
> > **Response to Reviewer 3YqS**
> >
> > We are immensely grateful for your thorough feedback on our manuscript. Your detailed feedback has significantly contributed to improving our work, and we deeply appreciate your valuable time and effort you have invested in this process.
> >
> >
> >
> > **Clarification on "Control" Term**. Thank you for your suggestion to provide a more detailed description of the "control" term in our paper. In response to your feedback, we are revising our manuscript to include a clearer explanation of our ODE equation, described as $\dot{X}=AX+u(t)$. Here, $u(t)$ represents the control term, which is specifically designed for fine-tuning the representations of vertices and hyperedges. We will further clarify that $u(t)=g(X(t))$, where $g$ function is implemented as a Feed-forward Network in our model. This explicit description aims to enhance understanding of how the control term operates within our model, providing the necessary adjustments to the diffusion process for more accurate and effective representation learning. We believe that this additional detail will help in clarifying any confusions regarding the role and functionality of the control term in our model.
> >
> >
> >
> > **Revising Terminology for Clarity**. We greatly appreciate your comments regarding the use of the term "controllability" in our manuscript. We understand the potential for misunderstanding given the term's strict definition in control theory. Your feedback has highlighted the importance of precise terminology, especially when integrating concepts from different fields such as computer science and control theory. We would try our best to finish carefully revising our manuscript to avoid any possible confusions. To this end, we are altering terminologies that accurately describe the capability of our hypergraph neural network model to be fine-tuned or controlled. We are dedicated to correcting any areas that might lead to different interpretations, ensuring that our work is both accurate and accessible to readers from various backgrounds.
> >
> >
> >
> > **Commitment to Stability Analysis.** Thank you for your constructive feedback regarding the stability analysis in our work, particularly in the context of feedback systems for nonlinear dynamical systems. We understand the importance of addressing this issue rigorously, especially given the complexity involved in such analyses. We acknowledge that our initial treatment of the control term and its impact on the system's stability may have been simplified. In response to your valuable insights, we are committed to conducting a more thorough and rigorous stability analysis of our ODE model. Specifically, we approach the control term $u(t)$ as a minor nonlinear perturbation, which remains governed by a suitable Lyapunov function. We will linearize $f$ and analyze the combined effect of this linearized part with the diffusion term. Our goal is to ensure that the eigenvalues of the resulting matrix lie in the left half of the complex plane, thereby guaranteeing the system's stability. That is, we investigate whether the remaining nonlinear parts of $f$ are controllable, with the intent that any higher-order terms in a Taylor expansion can be neglected. This approach will help us establish more concrete bounds for the control term $u(t)$, under which the system's stability can be assured. We will try our best to include these detailed stability results in the camera-ready version of our paper. This will involve carefully examining the constraints and bounds necessary to ensure the entire system's stability, thereby addressing your concerns in a more comprehensive and mathematically rigorous manner.
> >
> >
> >
> > We extend our gratitude once again for your feedback on our manuscript. Your guidance has been instrumental in steering us towards revisions that enhance the clarity and rigor of our work. We hope that our responses have addressed your concerns and we remain open to any further questions you might have.

---

> > > ### Comment · Reviewer_3YqS · 2023-11-22
> > >
> > > FYI, the description of the "control" term, you don't have to follow the path we gave (i.e. explicitly introducing u(t), which may not look as well as the math style you prefer). That is just an example (to describe what we think it about), which may deserve more consideration. You may find any better way to clarify this term in texts or math. Our comments just stress that more descriptions on this term in the paper will help a lot to avoid confusions.

---

> ### Comment · Reviewer_3YqS · 2023-11-22
>
> Thanks! Your current description on stability analysis sounds a reasonable path. Please ensure the stability analysis is rigorously addressed in your later version; even if it leads to sufficient conditions that are much stronger than the practical or what you expected (it is not surprising in theoretical analysis; the theory is usually much behind the practice), it is still better than leaving the current LTI theorem in the paper and it will certainly help readers to understand the stability issue.
>
> We have upgraded the score to 5. (If you had presented your update theorem here and it is correct, we would be very happy to upgrade to a higher score. Well we understand the difficulties behind and time is demanded, and we are sorry for feedback late and leaving you less time to make a fix during rebuttal.)

---

> > ### Author Response · Authors · 2023-11-22
> > **Response to Reviewer 3YqS**
> >
> > We deeply appreciate your feedback. We understand the importance of rigorous stability analysis in our work and take your suggestions seriously.
> >
> >
> >
> > Although the rebuttal period provides limited time, we try our best efforts to address your concerns regarding the stability analysis. We describe the stability analysis currently available. If we put the control term and the diffusion term all together, the equation is $\dot{X}=AX+\sigma(XW+b)$. Here, $\sigma$ is the activation function, specifically implemented as the ReLU function, and $W,b$ are the learnable parameters in the control term. In particular, we write $W$ as the right multiplication of $X$ because it is a transformation of features, which is distinguished from the diffusion effect among vertices and hyperedges in $AX$. As the property of $\sigma(\cdot)=\max(0,\cdot)$, implies that if the overall control term reaches zero, the system's overall stability reverts to that determined solely by the diffusion term.
> >
> >
> >
> > In cases corresponding to $\dot{X}=AX+XW+b$, we note the resemblance to a Sylvester System $\dot{X}=AX+XW$. The stability in this scenario hinges on two conditions, as demonstrated in prior works [1]. Firstly, $A,W$ must be continuous square matrices on $\mathbb{R}$. And secondly, there exists a positive constant $K$ such that $\int_{-\infty}^{\infty}\left\|\Phi(t) Y(t) P Z^*(t) \Psi^*(t)\right\| \leq K$, $\forall t \geq 0$, where $Y(t),Z(t)$ are fundamental matrix solutions of the linear system $\dot{X}=AX$ and $\dot{X}=XW$, respectively, with $\Phi(t), \Psi^*(t)$ being their corresponding bounded solutions. When these conditions are combined with $b=0$, the system maintains overall stability, albeit under strong constraints. We aim to identify broader conditions and bounds that assure system stability, addressing this challenge in a comprehensive and mathematically rigorous manner.
> >
> >
> >
> > We hope to provide a more detailed and thorough stability analysis in our final paper. In the coming period, we will dedicate more time to expanding and refining our results to present a more comprehensive understanding of the stability aspects of our model. We believe these additional efforts will not only strengthen our paper but also provide valuable insights for our readers.
> >
> >
> >
> > We are warmly open to further discussions on all aspects of our work to ensure its quality.
> >
> >
> >
> > [1] Kanuri, K. V. V., S. Bhagavathula, and K. Murty. "Stability Analysis of Linear Sylvester System of First Order Differential Equations." *International Journal of Engineering and Computer Science* 9.11 (2020).

---

> ### Comment · Reviewer_3YqS · 2023-11-23
>
> Thanks! Regarding your early-access updates on stability, please check the dimension-matching of matrix and vectors in $\dot{X} = AX + \sigma(XW + b)$. It is very good to take into account the specific tunable form of the control term; due to the specific form in NN, we don't have deal with complicated nonlinearity.  However, if we did not miss something, $X$ is the state variable right? How could these terms match their matrix-vector dimensions (for adding/multiplication), in $\dot{X} = AX + XW + b$ (I know your XW+b comes from NN, which seems not directly match the dimensions). It is not a big issue and can be easily fixed. And note that, if you have multiple-layer NN to implement the control term, the you cannot just analyze one layer.
>
> Overall we think you have been on the right way to discuss the stability. We would like to upgrade to 6, according to the preliminary results. Please make sure that the stability result will be properly updated in your final version.

---

> > ### Author Response · Authors · 2023-11-23
> > **Response to Reviewer 3YqS**
> >
> > We are immensely grateful for your review of our work and the encouragement that your revised score brings. Your constructive feedback has been instrumental in refining our approach and deepening our analysis.
> >
> >
> >
> > You are correct in pointing out that $X$ is the state variable in our model, with dimensions $|\mathcal{V}|+|\mathcal{E}|$ rows and $c$ columns, where $\mathcal{V}$ and $\mathcal{E}$ represent the sets of vertices and hyperedges, respectively, and $c$ denotes the feature dimension. Each row of $X$ corresponds to the feature of a vertex or a hyperedge. The matrix $A$ is a $(|\mathcal{V}|+|\mathcal{E}|)\times(|\mathcal{V}|+|\mathcal{E}|)$ square matrix. Regarding the matrix $W$, it is a $c\times c$ learnable square matrix. The product $XW$ thus signifies the transformation of each vertex/hyperedge representation across different feature channels.
> >
> >
> >
> > We understand that the term $b$, which is a $1\times c$ matrix (or can be understood as a transposed vector of dimension $c$), might have caused some confusion. In the context of neural networks, the expression $XW+b$ means that each row of  $XW$ is added with $b$. To illustrate, $(XW)\_{0,:}=(XW)\_{0,:}+ b$, $(XW)\_{1,:}=(XW)\_{1,:}+ b$, and so on.
> >
> >
> >
> > In our current implementation, the control term is realized through a single-layer neural network. Your suggestion is well-taken that if multiple layers are used to implement the control term in future work, a thorough reevaluation of the overall system's stability will be required.
> >
> >
> >
> > We are deeply thankful for your meticulous review and the valuable suggestions you have provided, which have significantly contributed to the improvement of our work. Your willingness to update the score is greatly appreciated and serves as a strong encouragement to our team. Thank you once again for your thoughtful feedback and support of our work.

---

### Author Response · Authors · 2023-11-20
**Invitation for Further Queries Regarding Our Work or Rebuttal**

Dear Reviewers,
$\newline$

We express our gratitude for your valuable time in reviewing our manuscript. We welcome any further questions you may have regarding our work or rebuttal. Please do not hesitate to share any additional concerns. We are always open to further discussion.
$\newline$

Sincerely,
$\newline$

The Authors.

---

### Author Response · Authors · 2023-11-23
**Summary of Discussion**

Dear Chairs and Reviewers,



I hope this message finds you well.



As the discussion period comes to a close, we would like to provide a concise summary of our interactions with the reviewers for your reference.



First and foremost, we extend our deepest gratitude to all the reviewers for their insightful and constructive feedback. It is encouraging that the reviews found our paper is

- R1: with good performance
- R2: a novel idea, a good attempt, and well organized and writing
- R3: well-written, easy to follow, and very complete
- R4: an interesting idea, solid and persuasive analysis



We read reviewer's comments carefully and tried our best to respond in detail. All of this will be addressed in the final version.



We summarize the main concerns of the reviewer and corresponding responses below.

- **Advantage of Combining Hypergraph and ODE**

  We clarify our model leverages the strengths of both hypergraph neural networks and graph ODE. Our model captures high-order correlation beyond pairwise interactions that are often overlooked in graph models. Unlike discrete models, such as HGNN, which can only capture representation at a few hidden layers, our ODE-based hypergraph method continuously tracks the evolution of these representations until finally stable. Besides, our model lies in its capacity to operate effectively at higher layer depths. And another key advantage of our model is its stability at increased layer depths.

- **Performance of the Model**

  We add an additional ablation study to further validate the performance of our model in scenarios with significantly few-shot conditions, especially datasets with only 5 or even 1 training vertex per class. Our model shows a more obvious advantage over general hypergraph neural network methods. This improvement becomes clearer in these few-shot scenarios. The reason for this can be attributed to the fact that with fewer training vertices, the need to capture information beyond 1-2 hops becomes more critical since fewer labels within the neighbor range with low distance. Our model's ability to leverage a broader scope of interaction allows it to access and utilize more comprehensive information, which is especially beneficial in situations where known vertex information is scarce.

- **Stable Analysis**

  We describe the stability analysis currently available. As the property of $\sigma(\cdot)=\max(0,\cdot)$, implies that if the overall control term reaches zero, the system's overall stability reverts to that determined solely by the diffusion term. Corresponding to another case, we note similarities with Sylvester's system. We presently give broad conditions for stability in this case, as demonstrated in previous work.



Based on the discussion with reviews, we also present a brief summary of our paper as follows.

- **Observation**: Existing hypergraph neural networks only tolerate small diffusion steps, and their performance drops significantly when the number of layers increases.
- **Solution**: We propose an ODE-based hypergraph framework that effectively increases the number of diffusion steps to capture long-range relations among vertices and still maintains high performance as the number of layers increases.
- **Results**: Extensive empirical evaluation on 9 real-world hypergraph datasets demonstrate the effectiveness of our framework. In addition, we also conduct experiments on datasets lacking initial vertex features or lacking initial hypergraph structure to demonstrate the effectiveness of our model in different scenarios.
- **Highlights**: We reference the concept of dynamical systems to enhance the existing hypergraph neural network, which characterizes dynamic continuous representations. We design a multi-layer framework as its neural implementation to generate accurate vertex representations. Our model achieves the best performance compared with all methods and achieves stable performance with respect to the increased layers.



Thanks again for your efforts in the reviewing and discussion. We appreciate all the valuable feedback that helped us to improve our submission.



Sincerely,



Authors of Paper 3379

---

### Meta-Review · Area_Chair_fYwX · 2023-12-14

**Metareview:**

This paper propose a new hypergraph neural networks (HGNNs) methodology through the lens of hypergraph dynamic systems (HDS). Based on the notion of  HDS, the authors propose a multi-layer HDS-ODE as a neural implementation of HDS. This new architecture improves the controllability and stability while it can capture long-range correlations. The proposed method is examined on several tasks where the proposed HGNN is shown to perform better than existing methods.

Overall this paper is well written. The motivation of the proposed method is well exposed and the properties of HDS are studied in a sufficient way. Moreover, the numerical experiments are through so that it is shown that the proposed method has good empirical performance in a convincing way.
Although there was some concern about the terminology of dynamical system such as "stability", the authors properly addressed the issue during the rebuttal.

In summary, this is a good quality paper which proposes a useful method. Then, I recommend acceptance to ICLR2024.

**Justification For Why Not Higher Score:**

Although the proposed method shows good performance empirically, some theoretical characterization of the method is not completely rigorous. If there were not such a (minor) concern, we could give higher score.

**Justification For Why Not Lower Score:**

One reviewer criticized the fact that this paper is not applied to a dynamical data. But, it is not a problem in this paper because the dynamical system point of view is applied to the layer direction, not the input data sequence direction. Hence, this paper is worth acceptance.

---

### Decision · Program_Chairs · 2024-01-16

Accept (poster)